# Generalization or Specificity? Spectral Meta Estimation and Ensemble (SMEE) with Domain-specific Experts

## Abstract

Existing domain generalization (DG) methodologies strive to construct a unified model trained on diverse source domains, with the goal of achieving robust performance on any unseen test domain. However, in practice, not all source domains contribute equally to effective knowledge transfer for a specific test domain. Consequently, the reliability of single-model generalization often falls short of classic empirical risk minimization (ERM). This paper departs from the conventional approaches and advocates for the paradigm that prioritizes specificity over broad generalization. We propose the Spectral Meta Estimation and Ensemble (SMEE) approach, which capitalizes on domain-specific expert models and leverages unsupervised ensemble learning to construct a weighted ensemble for test samples. Our comprehensive investigation reveals three key insights: (1) The proposed meta performance estimation strategy for model selection within the sources plays a pivotal role in accommodating stochasticity; (2) The proposed spectral unsupervised ensemble method for transferability estimation excels in constructing robust learners for multi-class classification tasks, while being entirely hyperparameter-free; and (3) Multi-expert test-time transferability estimation and ensemble proves to be a promising alternative to the prevailing single-model DG paradigm. Experiments conducted on the DomainBed benchmark substantiate the superiority of our approach, consistently surpassing state-of-the-art DG techniques. Importantly, our approach offers a noteworthy performance enhancement in online incremental testing while maintaining remarkable computational efficiency, executing in mere milliseconds per test sample during inference.

## 1 Introduction

The remarkable success of machine learning could not veil the excessive reliance on the independent and identically distributed (i.i.d.) assumption behind almost every algorithm. In practice, this assumption frequently falters, underscoring the critical importance of equipping models with the capacity to handle domain shifts (Zhou et al., 2023). Although scaling of data and model parameters has led to unprecedented performance in specific fields (Wei et al., 2022), models trained on limited data collections often suffer severe performance degradation when evaluated on out-of-distribution test samples, across a spectrum of fields (Recht et al., 2019). In stark contrast, human adaptability to novel environments or tasks with minimal effort and time raises profound questions regarding the nature of this adaptability, whether it arises from the cumulative acquisition of wisdom or the ability to correlate current tasks with existing knowledge.

The current endeavor on domain generalization (DG) faces a similar question. DG considers utilizing multiple training domains to enhance the out-of-distribution generalization ability of models. In most works, the practice is to find a single optimal model learned on multiple training domains, assuming that such a model encapsulates the collective knowledge necessary to address any distributional shift within an unseen target domain. Nonetheless, the state-of-the-art DG approaches have demonstrated unstable performance, frequently oscillating below the classic empirical risk minimization (ERM) (Gulrajani & Lopez-Paz, 2021). Furthermore, recent investigations have emphasized the limitations inherent in the reliance on a solitary model or specific architectural configurations (Li et al., 2023c), while ensemble approaches have exhibited enhanced stability and performance in

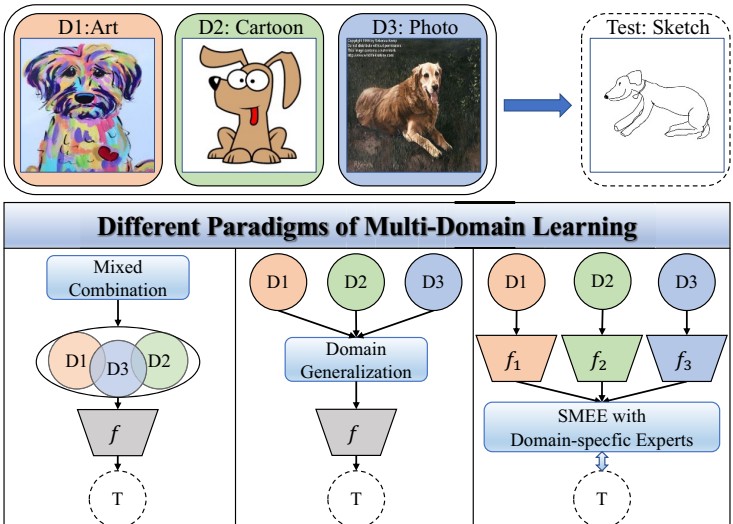

Figure 1: Three paradigms to approach multi-domain learning. DG considers using domain-wise datasets instead of a simple mixed combination of data to utilize the domain information. We propose to use domain-specific experts to strengthen specificity instead of generalization. SMEE aggregates predictions of experts using meta supervised and spectral unsupervised performance estimation approaches.

recent works (Zhou et al., 2021; Cha et al., 2021; Arpit et al., 2022). Additionally, recognizing the indispensability of specific test domain information for precise estimation of transferability (Jiang et al., 2022) and adaptivity gap (Dubey et al., 2021) remains paramount to mitigating the risks associated with negative transfer (Zhang et al., 2023a).

In light of these considerations, we reevaluate and challenge the prevailing DG paradigm, characterized by (1) the reliance on a single model to learn across multiple training domains, (2) the assumption of equal transferability among all source domains to the target domain, and (3) the prohibition of access to the target domain. The inherent limitations of single-model DG become strikingly evident when the source domains are diverse since each source domain contains much domain-specific information potentially useful for a target domain, which should be retained (Zhou et al., 2021). As for the target domain, it is usually prohibited from access in DG for consideration of real-world applications. Nevertheless, recent advances in test-time adaptation (Liang et al., 2023) shed light on the feasibility of utilizing information from test data in real-time, aligning better with practical deployment scenarios.

Inspired by attempts in recent research, we argue that single-model DG is over-optimistic given the wide range of real-world applications. In fact, previous works have already offered enough emphasis on such paradigm with the Mixture-of-Expert (MoE) models (Zhong et al., 2022). In this work, we consider using individual ERM models for each source. To facilitate source model selection, we introduce a meta performance estimation technique for model selection within the sources. For the test domain, we propose an approach based on spectral unsupervised ensemble learning to assess the transferability of each source model to test samples. Importantly, our approach leverages model predictions without necessitating re-training, adaptation, or iterative optimization, offering a simplified, more stable, and computationally efficient alternative compared to existing DG strategies. A comparison of the learning paradigms is shown in Figure 1.

To summarize, the major contributions of this work are as follows:

1. Our work extends the Spectral Meta-Learner (SML) (Parisi et al., 2014) unsupervised ensemble learning approach from binary to multi-class classification, showcasing superior performance over uniform-weights averaging and majority voting with minimal computational overhead.

2. We demonstrate that domain-specific experts in multi-domain learning can be enhanced using a straightforward yet remarkably effective meta model selection approach.

3. Our proposed test-time transferability estimation and ensemble approach follows a more practical alternative to the prevailing single-model DG paradigm. Comprehensive experiments on DG benchmark datasets illustrate its promise in prioritizing specificity over generalization.

## 2 RETHINKING SINGLE-MODEL DOMAIN GENERALIZATION

DG approaches usually aim for learning domain-invariant representations (Albuquerque et al., 2019). Such approaches have been pointed out to heavily rely on utilizing the unsupervised test domain information for good performance (Li et al., 2018c). We empirically substantiate such point in the following discussion. Visualizations of feature distribution across domains under domain-invariant representation learning, and the performance comparison of DG ERM and domain-specific ERM are shown in Figure 2-3.

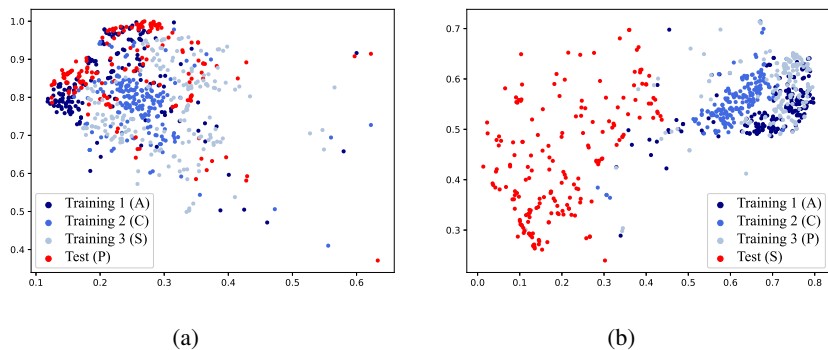

(a)                                    (b)

Figure 2: Visualizations using PACS (Li et al., 2017) dataset where the four domains are (P)hoto, (A)rt, (C)artoon, and (S)ketch. t-SNE (Van der Maaten & Hinton, 2008) visualization of deep representations of samples in the class 'Dog' using DANN (Ganin et al., 2016) for DG approach are shown when the test domain is (a) (P)hoto; and (b) (S)ketch. More illustrations can be found in Appendix A.2. Best viewed in color.

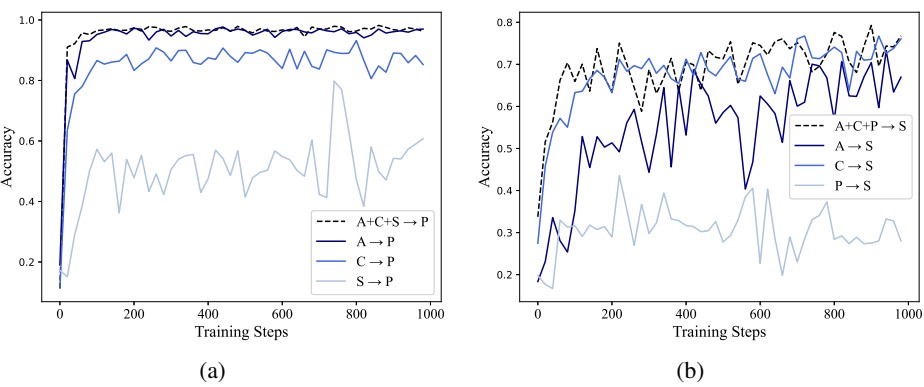

(a)                                    (b)

Figure 3: Test performance of DG ERM and domain-specific ERM when the test domain is (c) (P)hoto; and (d) (S)ketch. More illustrations can be found in Appendix A.2. Best viewed in color.

Observations yield the following insights:

1. The importance of specific test domain information. Although domain-invariant feature representation learning could minimize the average risk across the training domains (Muandet

et al., 2013), the test domain distribution is intractable. It may seem that domain-invariant representation is achieved as in Figure 2(a). However, in Figure 2(b), there are still significant disparities between the test domain (S)ketch distribution and training domain distributions.

2. Distinct adaptivity gap (Dubey et al., 2021; Zhang et al., 2023b). As in Figure 3(a) and Figure 3(b), the performance of transferring from (A)rt $\mapsto$ (P)hoto and (C)artoon $\mapsto$ (S)ketch is much higher than the other transfer routes with a single source domain.

3. Uncertainty surrounding generalization mechanism. Notably, as in Figure 3(a) and Figure 3(b), performance of transferring from (A)rt $\mapsto$ (P)hoto almost matches with (A)rt+(C)artoon+(S)ketch $\mapsto$ (P)hoto, and (C)artoon $\mapsto$ (S)ketch almost matches with (A)rt+(C)artoon+(P)hoto $\mapsto$ (S)ketch. This observation raises the question of whether DG achieves generalization through domain-invariant representation learning or if it excels only when a training domain exhibits high transferability to the test domain. Furthermore, as in Figure 2(b), (C)artoon has much higher transferability to (S)ketch compared to the other two sources as in Figure 3(b). It prompts us to consider whether learning invariant representations within the sources would even downgrade transfer performance.

These insights collectively highlight a crucial point: the transferability of any source domain remains unpredictable without access to the specific test domain information, leading to the inherent instability witnessed in current DG approaches. Then why not use domain-specific experts to prioritize specificity over generalization? Our proposed paradigm offers the advantage of explicit knowledge sourcing for each expert. Consequently, it becomes feasible to analyze the expert transferability during test phase using target domain information, presumably in a simple, online, and computationally efficient way. The goal, therefore, revolves around determining the transferability of source domains or models at test phase for a presumably better aggregation than conventional uniform-weights averaging or majority voting. The question that remains to be left is: how to reliably evaluate the transferability of multiple source experts and identify an optimal aggregation strategy in a strictly unsupervised manner, given an unknown level of distributional shift within the test domain?

## 3 SPECTRAL META ESTIMATION AND ENSEMBLE (SMEE)

### 3.1 PROBLEM SETUP

Assume that training data from $S$ source domains $\{\mathcal{D}_s^i\}_{i=1}^S$ are available. Each source domain is composed of $n_i$ labeled samples $\mathcal{D}_s^i = \{(\mathbf{x}_l^i, y_l^i)\}_{l=1}^{n_i}$, where $\mathbf{x} \in \mathcal{X} \subset \mathbb{R}^d$, $y \in \mathcal{Y} = \{1, 2, ..., K\}$ consists of $K$ classes, which are sampled from distinct distributions $\mathcal{D}_s^i \sim \mathcal{P}_{XY}^i$, where $P_{XY}$ denotes the joint distribution of the input sample and output label, and $\mathcal{P}_{XY}^i \neq \mathcal{P}_{XY}^{i'}$ for $1 \leq i \neq i' \leq S$. The goal is to learn from $\{\mathcal{D}_s^i\}_{i=1}^S$ to achieve good performance on an unseen out-of-distribution target domain $\mathcal{D}_t = \{(\mathbf{x}_i, y_i)\}_{i=1}^{n_t} \sim \mathcal{P}_{XY}^t$, where $y_i$ are the true labels, and $\mathbf{x} \in \mathcal{X} \subset \mathbb{R}^d$, $y \in \mathcal{Y} = \{1, 2, ..., K\}$ but $\mathcal{P}_{XY}^t \neq \mathcal{P}_{XY}^i$ for $1 \leq i \leq S$, i.e., the target domain samples are drawn from a distinct distribution from the source domains but have identical label spaces.

DG aims to train a single generalized model on the entire source domains, i.e., $f(\{\mathcal{D}_s^i\}_{i=1}^S; \boldsymbol{\theta})$. In this work, we propose to train domain-specific models for each source domain, i.e., $\{f(\mathcal{D}_s^i; \boldsymbol{\theta}_i)\}_{i=1}^S$, and aggregate their predictions on target domain $\mathcal{D}_t$ at test time.

### 3.2 UNSUPERVISED PERFORMANCE EVALUATION WITH MULTI-CLASS SML

Consider $M \geq 3$ predictors $\{f_j\}_{j=1}^M$ are available to make predictions on test samples $\mathcal{D}_t = \{(\mathbf{x}_i)\}_{i=1}^{n_t}$. The goal is to utilize the predictions $f_j(\mathbf{x}_i)$ of the predictors and reliably rank their performance without resorting to the true labels. More preferably, an ensemble learner could be built that can achieve better classification results than uniform-weights averaging or majority voting using $f_j(\mathbf{x}_i)$.

The Spectral Meta-Learner (SML) (Parisi et al., 2014) offers such an ensemble strategy, with preliminaries can be found in Appendix A.3. We extend it to multi-class classification with $K > 2$. We use the one-vs-rest paradigm for solving multi-class problem based on a binary setup. Specifically, we consider $K$ splits of $\mathcal{A}_k \cup (\mathcal{Y} \setminus \mathcal{A}_k)$ and $\mathcal{A}_k = \{k\}$. On test samples, a predictor would return

a vector of probability $f_j(\mathbf{x}_i)$ over the $K$ classes where $\delta_k\big(f_j(\mathbf{x}_i)\big)$ denotes the $k$-th probability corresponding to the $k$-th class. The predictions are first converted from soft to hard labels and then converted to binary for $\mathcal{A}_k \cup (\mathcal{Y} \setminus \mathcal{A}_k)$, i.e., $f_j(\mathbf{x}_i)_{(k)} = 1$ if $\arg\max_k \delta_k\big(f_j(\mathbf{x}_i)\big) = k$, or $-1$ otherwise to indicate positive or negative prediction.

With a few assumptions below, the following lemma, proved in Appendix A.4, illustrates that applying the SML ensemble to a multi-class classification task is applicable.

1. Test data $\mathcal{D}_t = \{(\mathbf{x}_i)\}_{i=1}^{n_t}$ satisfy the i.i.d. assumption.
2. Conditional independence between predictors $\{f_j\}_{j=1}^{M}$.
3. Most predictors have performance better than random.
4. Test data $\mathcal{D}_t = \{(\mathbf{x}_i)\}_{i=1}^{n_t}$ are class-balanced.

**Lemma 1.** *(Multi-Class SML) Consider all one-vs-rest splits of classes over all $K$ classes where each split is $\mathcal{A}_k \cup (\mathcal{Y} \setminus \mathcal{A}_k)$ that $\mathcal{A}_k = \{k\}$. Using $\mathcal{Q}_{(k)}$, the $M \times M$ population covariance matrix using binary class prediction $f_j(\mathbf{x}_i)_{(k)}$ of the $M$ predictors, $\mathcal{Q}_{(k)}$ would have off-diagonal entries identical to that of a rank-one matrix $R_{(k)} = \lambda \boldsymbol{v}_{(k)} \boldsymbol{v}_{(k)}^\top$ with eigenvector $\boldsymbol{v}_{(k)}$ and eigenvalue $\lambda$. The entries $\bar{v}_j$ of the averaged eigenvectors $\bar{\boldsymbol{v}} = \frac{1}{K}\sum_{k=1}^{K} \boldsymbol{v}_{(k)}$ would be proportional to the $\hat{\pi}$-performance of the $M$ predictors $\{f_j\}_{j=1}^{M}$.*

The $\hat{\pi}_j$-performance refers to $\hat{\pi}_{j(k)} = \frac{1}{2}(P[f_j(\mathbf{x}) = k | Y = k] + P[f_j(\mathbf{x}) \neq k | Y \neq k])$ of predictor $f_j$ in class split of $\mathcal{A}_k \cup (\mathcal{Y} \setminus \mathcal{A}_k)$, where $b = P[Y = k] - P[Y \neq k]$ is the class imbalance, $\boldsymbol{v}_{(k)}$ can be estimated using approach described in Appendix A.3. Note that the $\hat{\pi}$-performance is equivalent to the macro balanced classification accuracy in the binary case.

Intuitively, SML utilizes inter-predictor correlations to figure out which predictors make similar and more credible results. Based on *Lemma* 1, we could obtain entries proportional to the estimated performances of predictors in a completely unsupervised manner using their predictions. Importantly, building on top of *Lemma* 1, we propose multi-class SML for unsupervised ensemble learning:

$$\hat{y}_i^{\text{Multi-Class SML}} = \arg\max_k \left( \sum_{j=1}^{M} \delta_k\big(f_j(\mathbf{x}_i)\big) \cdot \bar{v}_j \right). \tag{1}$$

The multi-class SML ensemble is expected to perform better than uniform-weights averaging or majority voting, as it assign greater weight to superior predictors exhibiting higher transferability to the test domain. Note that neural networks produced probability outputs instead of class membership. Our approach first used converted class memberships of predictions to calculate the weights in the ensemble and then applied the weights to probabilities to make the final ensemble inferences.

### 3.3 META MODEL SELECTION

It is essential to highlight that Assumption 3 in Section 3.2 requires most if not all of predictors in the ensemble to perform better than random. Therefore, aside from obtaining better ensemble predictions at test phase, it becomes imperative to filter out comparatively underperforming models within the ensemble during the training of source models. In this subsection, we explore the straightforward yet effective approach for evaluating the performance of domain-specific experts, employing a meta cross-validation paradigm. Following the recommendations outlined in Gulrajani & Lopez-Paz (2021) that '*A DG algorithm should be responsible for specifying a model selection method,*' a robust model selection approach is of vital significance within the context of our proposed paradigm centered around domain-specific ERM experts.

We propose a simple and intuitive strategy of using other source domains to evaluate the current source domain model. Assume that multiple ERM models are trained on one specific domain, i.e., $B$ models for source domain $i$ that $\{f^b(\mathcal{D}_s^i; \boldsymbol{\theta}_i^b)\}_{b=1}^{B}$ with different initialization and stochastic optimization process. We meta-select the best-performing models out of $\{f^b(\mathcal{D}_s^i; \boldsymbol{\theta}_i^b)\}_{b=1}^{B}$ which have the highest average accuracies on the other sources domains $\{D_s^{i'} | i' \neq i\}_{i'=1}^{S}$. In this way, the instability of stochastic optimization can be accommodated, and we are able to filter out generally worse-performing models.

## 3.4 SMEE WITH DOMAIN-SPECIFIC EXPERTS FOR DG

Our proposed ensemble learner only requires predictions of models on the test domain to achieve. However, one common requirement in DG is that the test domain information should not be used during training. Such concern is for deployment into real-world applications, where test samples are unknown beforehand. Although our approach is based on test-time aggregation of multiple experts, it also fits with real-world applications in the sense that:

1. Preservation of test information. No test data has been employed in constructing the underlying source models, preserving the integrity of the DG framework.

2. Minimal computational overhead at test phase. During test phase, our approach incurs minimal computational cost. It entails no model re-training, adaptation, or iterative optimization, making it amenable to online incremental settings where test samples arrive sequentially and each test sample queries for a recalculation of the covariance matrix using the cumulative test set until this exact test query.

3. Sole reliance on model predictions at test phase. It broadens its applicability to a diverse range of real-world use cases, including but not limited to: domain-incremental settings, where new training domains are introduced progressively; privacy-preserving scenarios, where models are only accessible via APIs presumably from other service providers offering inferences on queries while protecting model parameters.

These inherent characteristics position our proposed method as a versatile and practical solution for multi-domain learning, emphasizing compliance with rigorous DG requirements and adaptability to real-world settings. The pseudo-code of SMEE for online inference is given in Algorithm 1. An empirical analysis on comparing the online and offline inference can be found in Appendix A.6.

---

**Algorithm 1** Spectral Meta Estimation and Ensemble (SMEE).

---

**Input:** $S$ source domains $\{\mathcal{D}_s^i\}_{i=1}^S$, each with $n_i$ labeled samples $\mathcal{D}_s^i = \{(\mathbf{x}_l^i, y_l^i)\}_{l=1}^{n_i}$;
  Streaming target domain data $\mathcal{D}_t = \{(\mathbf{x}_i)\}_{i=1}^{n_t}$;
  $B$, the number of distinct models trained for each source domain;
  $M$, the total number of total models been kept after meta ranking from all source domains;
**Output:** The classification $\{\hat{y}_i\}_{i=1}^{n_t}$ for $\mathcal{D}_t = \{(\mathbf{x}_i)\}_{i=1}^{n_t}$.
  *// Meta Model Selection*
  **for** $i = 1 : S$ **do**
    Train $B$ domain-specific models for source domain $i$ that $\{f^b(\mathcal{D}_s^i; \boldsymbol{\theta}_i^b)\}_{b=1}^B$;
    Test and average performance metric of $\{f^b(\mathcal{D}_s^i; \boldsymbol{\theta}_i^b)\}_{b=1}^B$ on each of $\{D_s^{i'} | i' \neq i\}_{i'=1}^S$;
    Rank and select the best performing $\frac{M}{S}$ models $\{f^b(\mathcal{D}_s^i; \boldsymbol{\theta}_i^b)\}_{b=1}^{\frac{M}{S}}$;
  **end for**
  Assemble all the selected models as $\{f_j\}_{j=1}^M$;
  *// Online Inference*
  **for** $i = 1 : n_t$ **do**
    **if** $i \leq M$ **then**
      *// Resort to uniform-weights averaging when covariance matrix calculation is inadequate*
      Calculate $\hat{y}_i = \arg\max_k \sum_{j=1}^M \delta_k(f_j(\mathbf{x}_i))$;
    **else**
      *// Spectral Meta-Ensemble*
      **for** $k = 1 : K$ **do**
        Convert probability vector $f_j(\mathbf{x}_i)$ to binary $f_j(\mathbf{x}_i)_{(k)} \in \{1, -1\}$;
        Compute $\mathcal{Q}_{(k)}$, where $q_{ij(k)} = \mathbb{E}[(f_i(\mathbf{x})_{(k)} - \mathbb{E}[(f_i(\mathbf{x})_{(k)}])(f_j(\mathbf{x})_{(k)} - \mathbb{E}[(f_j(\mathbf{x})_{(k)}])]$;
        Calculate Singular Value Decomposition (SVD) of $\mathcal{Q}_{(k)}$ for its principal eigenvector $\boldsymbol{v}_{(k)}$;
      **end for**
      Compute entries $\{\bar{v}_j\}_{j=1}^M$ of $\bar{\boldsymbol{v}}$ by averaging the eigenvectors $\bar{\boldsymbol{v}} = \frac{1}{K} \sum_{k=1}^K \boldsymbol{v}_{(k)}$;
      Calculate $\hat{y}_i$ using Equation (1);
    **end if**
  **end for**

---

# 4 EXPERIMENTS

## 4.1 DATASETS AND ALGORITHMS

This section presents the evaluation results on DomainBed (Gulrajani & Lopez-Paz, 2021) DG benchmark including five datasets, namely PACS (Li et al., 2017), VLCS (Fang et al., 2013), OfficeHome (Venkateswara et al., 2017), TerraIncognita (Beery et al., 2018), and DomainNet (Peng et al., 2019). Their statistics and examples can be found in Appendix A.5.

We compared with the classic ERM, and state-of-the-art DG algorithms that perform better than ERM on average on the five datasets, including Marginal Transfer Learning (MTL) (Blanchard et al., 2021), Mixup (Xu et al., 2020), Meta-Learning for Domain Generalization (MLDG) (Li et al., 2018a), Fish (Shi et al., 2022), and its variant Fishr (Rame et al., 2022), CORrelation ALignment (CORAL) (Sun et al., 2016), meta-Domain Specific-Domain Invariant (mDSDI) (Bui et al., 2021), Style-agnostic Networks (SagNets) (Nam et al., 2019), test-time Adaptation with Non-Parametric Classifier (AdaNPC) (Zhang et al., 2023c), Meta-Distillation of Mixture-of-Experts (Meta-DMoE) (Zhong et al., 2022), Domain-specific Risk Minimization (DRM) (Zhang et al., 2023b), Stochastic Weight Averaging Densely (SWAD) (Cha et al., 2021), Generalizable Mixture-of-Experts (GMoE) (Li et al., 2023a), Ensemble of Averages (EoA) (Arpit et al., 2022), Mutual Information Regularization with Oracle (MIRO) (Cha et al., 2022), SpecIalized Model-samPLE matching (SIMPLE) (Li et al., 2023c).

## 4.2 NECESSITY OF META MODEL SELECTION BEFORE ENSEMBLE

Utilizing our straightforward yet effective meta model selection approach, which entails selecting the single top-performing expert from a modest pool of ten randomly initialized ERM models for each source domain, we achieve performance parity with the multi-source combined DG ERM approach. The results are shown in Figure 4. Observe that meta model selection substantially improves over the random model selection, suggesting the necessity to suppress the randomness of stochastic optimization with domain-specific experts.

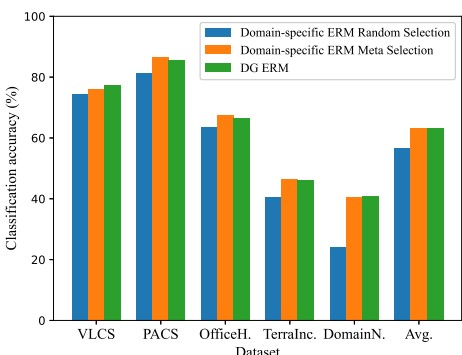

Figure 4: Performance comparison of domain-specific ERM and DG ERM with meta or random model selection on five datasets.

## 4.3 EVALUATION RESULTS ON DOMAINBED BENCHMARK

The main results of average classification accuracies using the leave-one-domain-out evaluation paradigm are shown in Table 1. Since SML requires a decent amount of predictors for covariance matrix calculation, all SMEE ensembles use five models from each source. All used models are the standard ResNet-50 model pre-trained on ImageNet, except for the model-zoo comparison with SIM-PLE using a combination of five different architectures (ResNet-50, EfficientNet_b0, DenseNet201, ViT_b_16, ShuffleNet_v2_x1_0) also pre-trained on ImageNet. SMEE algorithms used meta model selection to select five out of ten randomly initialized independently trained models for each source domain. All experiments were repeated ten times and the average results are reported.

Table 1: Average classification accuracies on five datasets. Baseline results are from either original publications or reproduced by DomainBed (Gulrajani & Lopez-Paz, 2021), under the same setup. The bracket in model zoo ensemble indicates the number of models used in the ensemble.

| Algorithm | VLCS | PACS | OfficeH. | TerraInc. | DomainN. | Avg. |
|---|---|---|---|---|---|---|
| *Non-ensemble Algorithms* | | | | | | |
| ERM (Gulrajani & Lopez-Paz, 2021) | $77.5_{\pm0.4}$ | $85.5_{\pm0.2}$ | $66.5_{\pm0.3}$ | $46.1_{\pm1.8}$ | $40.9_{\pm0.1}$ | 63.3 |
| MTL (Blanchard et al., 2021) | $77.2_{\pm0.4}$ | $84.6_{\pm0.5}$ | $66.4_{\pm0.5}$ | $45.6_{\pm1.2}$ | $40.6_{\pm0.1}$ | 63.4 |
| Mixup (Xu et al., 2020) | $77.4_{\pm0.6}$ | $84.6_{\pm0.6}$ | $68.1_{\pm0.3}$ | $47.9_{\pm0.8}$ | $39.2_{\pm0.1}$ | 63.4 |
| MLDG (Li et al., 2018a) | $77.2_{\pm0.4}$ | $84.9_{\pm1.0}$ | $66.8_{\pm0.6}$ | $47.7_{\pm0.9}$ | $41.2_{\pm0.1}$ | 63.6 |
| Fish (Shi et al., 2022) | $77.8_{\pm0.3}$ | $85.5_{\pm0.3}$ | $68.6_{\pm0.4}$ | $45.1_{\pm1.3}$ | $42.7_{\pm0.2}$ | 63.9 |
| CORAL (Sun et al., 2016) | $78.8_{\pm0.6}$ | $86.2_{\pm0.3}$ | $68.7_{\pm0.3}$ | $47.6_{\pm1.0}$ | $41.5_{\pm0.1}$ | 64.6 |
| mDSDI (Bui et al., 2021) | $79.0_{\pm0.3}$ | $86.2_{\pm0.2}$ | $69.2_{\pm0.4}$ | $48.1_{\pm1.4}$ | $42.8_{\pm0.1}$ | 64.6 |
| SagNet (Nam et al., 2019) | $77.8_{\pm0.5}$ | $86.3_{\pm0.2}$ | $68.1_{\pm0.1}$ | $48.6_{\pm1.0}$ | $40.3_{\pm0.1}$ | 64.7 |
| Fishr (Rame et al., 2022) | $78.2_{\pm0.2}$ | $86.9_{\pm0.2}$ | $68.2_{\pm0.2}$ | $53.6_{\pm0.4}$ | $41.8_{\pm0.2}$ | 65.7 |
| MIRO (Cha et al., 2022) | $79.0_{\pm0.0}$ | $85.4_{\pm0.4}$ | $70.5_{\pm0.4}$ | $50.4_{\pm1.1}$ | $44.3_{\pm0.2}$ | 65.9 |
| AdaNPC (Zhang et al., 2023c) | $79.5_{\pm2.4}$ | $88.8_{\pm0.1}$ | / | $53.9_{\pm0.3}$ | $42.9_{\pm0.5}$ | / |
| *Ensemble Algorithms* | | | | | | |
| Meta-DMoE (Zhong et al., 2022) | / | 86.9 | / | / | 44.2 | / |
| DRM (Zhang et al., 2023b) | $\underline{80.5}_{\pm0.3}$ | $88.5_{\pm1.2}$ | / | / | $42.4_{\pm0.1}$ | / |
| SWAD (Cha et al., 2021) | $79.1_{\pm0.1}$ | $88.1_{\pm0.1}$ | $70.6_{\pm0.2}$ | $50.0_{\pm0.3}$ | $46.5_{\pm0.1}$ | 66.9 |
| GMoE (Li et al., 2023a) | $80.2_{\pm0.2}$ | $88.1_{\pm0.1}$ | $\mathbf{74.2}_{\pm0.4}$ | $48.5_{\pm0.1}$ | $\mathbf{48.7}_{\pm0.2}$ | 67.9 |
| EoA (Arpit et al., 2022) | 79.1 | $\underline{88.6}$ | 72.5 | 52.3 | 47.4 | 68.0 |
| MIRO + SWAD (Cha et al., 2022) | $79.6_{\pm0.2}$ | $88.4_{\pm0.1}$ | $72.4_{\pm0.1}$ | $\mathbf{52.9}_{\pm0.2}$ | $47.0_{\pm0.0}$ | $\underline{68.1}$ |
| SMEE (ours) | $\mathbf{82.1}_{\pm0.4}$ | $\mathbf{90.4}_{\pm0.4}$ | $\underline{73.2}_{\pm0.6}$ | $\underline{52.8}_{\pm2.1}$ | $48.1_{\pm0.6}$ | $\mathbf{69.3}$ |
| *Model Zoo Ensemble* | | | | | | |
| SIMPLE (15) (Li et al., 2023c) | $79.8_{\pm0.1}$ | $84.1_{\pm0.5}$ | $79.9_{\pm0.1}$ | $56.8_{\pm0.2}$ | $46.3_{\pm0.4}$ | 69.4 |
| SIMPLE (224) (Li et al., 2023c) | $79.9_{\pm0.5}$ | $88.6_{\pm0.4}$ | $\mathbf{84.6}_{\pm0.5}$ | $\mathbf{57.6}_{\pm0.8}$ | $\mathbf{49.2}_{\pm1.1}$ | $\mathbf{72.0}$ |
| SMEE (15-25) (ours) | $\mathbf{84.2}_{\pm0.6}$ | $\mathbf{91.6}_{\pm0.6}$ | $78.4_{\pm0.3}$ | $55.4_{\pm1.6}$ | $48.7_{\pm0.6}$ | 71.7 |

The main results are shown in Table 1, and more extended result can be found in Appendix A.9. Observe that ensemble algorithms generally perform better than non-ensemble ones. Our proposed approach, SMEE, outperforms all other approaches when limited to ResNet-50 architecture. Using a model zoo of multiple architectures, SMEE with 15-25 models (varied due to the different number of domains based on the dataset) almost matches the performance of SIMPLE using 224 models.

In Figure 5, we also show the performance of SMEE as the number of used models increases using meta or random model selection out of a pool of ten ResNet-50 models. Observe that the performance first increases sharply and then gains slowly. The lately added models into the ensemble have lower ranks in the meta model selection, indicating lower estimated performance and worse transferability, and do not help with performance improvement anymore. It also indicates that meta model selection can drastically improve performance when the number of models in the ensemble is relatively small.

## 4.4 TEST-TIME ENSEMBLE

For the proposed paradigm of using domain-specific experts for the ensemble on the test domain, we compared with other test-time ensemble approaches, also using five models per source domain for more stabilized results, selected using meta model selection from a pool of ten randomly initialized ResNet-50 models. In Table 2, we show the average single model performance in the ensemble, comparing with uniform-weights averaging, majority voting, and recent approaches based on neural networks. Prediction Entropy Measurement (PEM) (Zhang et al., 2023b) calculates entropy $H$ on prediction probabilities and uses the normalized $H^{-2}$ as transferability measurement. Prediction Uncertainty Measurement (PUM) variant uses top-1 to top-2 probability difference instead of entropy.

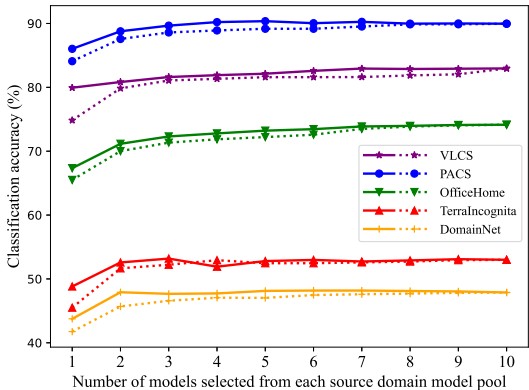

Figure 5: Performance of SMEE using meta (solid lines) or random (dashed lines) model selection from a pool of ten randomly initialized domain-specific ERM ResNet-50 models for each source.

Similarity measurements such as Cosine Similarity Measurement (CSM) and $l_2$ norm Similarity Measurement (L2SM) calculate the distance from the test sample representation to the average representations from each source domain (Zhang et al., 2023b), also using the normalized inverse square of the distance. SML-soft (Li et al., 2023b) uses soft probability scores to replace the $\{1, -1\}$ binary labels in SML covariance matrix calculation.

From the results, we found that distance measurement-based approaches could improve over uniform-weights averaging by a small margin, while uncertainty-based approaches did not seem to offer improvements. Our proposed multi-class SML outperforms uniform-weights averaging and majority voting, and also other baselines. Our two step convert-and-ensemble approach performs better and more stable than directly using probability scores for covariance matrix calculation as in SML-soft (Li et al., 2023b). Notably, the performance boost of our approach only requires milliseconds on average to execute for each test sample in online incremental setting, and is also free of hyperparameters. A complete discussion on computational cost can be found in Appendix A.8.

Table 2: Comparison of test-time transferability estimation and ensemble using domain-specific experts approaches. Average classification accuracies are shown, using five domain-specific experts meta-selected from each source of ten randomly initialized ResNet-50 models.

| Algorithm | VLCS | PACS | OfficeHome | TerraIncognita | DomainNet | Avg. |
|---|---|---|---|---|---|---|
| Single | $70.0_{\pm 5.2}$ | $71.0_{\pm 4.1}$ | $54.9_{\pm 4.8}$ | $40.6_{\pm 4.3}$ | $27.8_{\pm 6.2}$ | 52.9 |
| Voting | $80.1_{\pm 0.8}$ | $87.8_{\pm 0.7}$ | $72.2_{\pm 0.9}$ | $48.7_{\pm 2.9}$ | $45.6_{\pm 1.6}$ | 66.9 |
| Averaging | $80.3_{\pm 0.6}$ | $88.6_{\pm 0.5}$ | $72.9_{\pm 0.5}$ | $50.2_{\pm 2.4}$ | $46.9_{\pm 1.0}$ | 67.8 |
| PEM | $78.9_{\pm 1.0}$ | $88.7_{\pm 0.8}$ | $69.9_{\pm 2.3}$ | $50.0_{\pm 1.4}$ | $44.4_{\pm 1.2}$ | 66.4 |
| PUM | $80.4_{\pm 1.1}$ | $88.1_{\pm 1.1}$ | $72.8_{\pm 1.9}$ | $50.5_{\pm 1.6}$ | $46.5_{\pm 1.0}$ | 67.7 |
| CSM | $81.7_{\pm 0.6}$ | $88.3_{\pm 1.0}$ | $72.5_{\pm 0.6}$ | $51.0_{\pm 2.1}$ | $46.4_{\pm 1.2}$ | 68.0 |
| L2SM | $81.4_{\pm 0.7}$ | $89.3_{\pm 0.9}$ | $72.5_{\pm 0.5}$ | $50.9_{\pm 2.3}$ | $46.1_{\pm 1.1}$ | 68.4 |
| SML-soft | $\mathbf{82.5}_{\pm 0.6}$ | $85.0_{\pm 0.8}$ | $72.9_{\pm 1.1}$ | $51.9_{\pm 1.7}$ | $41.5_{\pm 1.3}$ | 66.8 |
| SMEE (ours) | $82.1_{\pm 0.4}$ | $\mathbf{90.4}_{\pm 0.4}$ | $\mathbf{73.2}_{\pm 0.6}$ | $\mathbf{52.8}_{\pm 2.1}$ | $\mathbf{48.2}_{\pm 0.6}$ | $\mathbf{69.3}$ |

## 5 CONCLUSION

In this work, we extend SML to the multi-class class, showcasing its effectiveness with contemporary neural network transfer in multi-domain learning. Besides comparing with the popular DG or transferability estimation approaches, we propose a ranking and ensemble approach for multiple predictors within the multiple sources and on the test domain. With our pioneering approach and its empirical success, we expect future work to emphasize more on domain-specific expertise for the myriad challenges posed by multi-domain learning scenarios.

## 6 Reproducibility Statement

To ease the reproduction efforts and improve reuse of the proposed approach, we provide multi-class SML as an easy-to-use one-python-file demo of APIs with minimal dependencies that simulate scikit-learn functions. The functions take two arguments of prediction of multiple neural network models on the test set and the true labels, and return the final evaluation scores. We also provide files of outputs of neural networks in the SMEE ensemble, which are trained on PACS dataset regarding Sketch as the test domain, as a small demo test suite. Our entire repository will be polished and uploaded to GitHub, containing all codes used in this paper.

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

# A   APPENDIX

## A.1   RELATED WORK

### A.1.1   DOMAIN GENERALIZATION

DG theory considers classifier $h$ for each of the $M$ source domains which contain $n$ samples each, and measures the average risk $\xi(h)$ over all possible target domains, of error bound proved in the following theorem.

**Theorem 2.** *(Average risk estimation error bound for binary classification (Blanchard et al., 2021)). Assume that the loss function $l$ is $L_l$-Lipschitz in its first argument and is bounded by $B_l$. Assume also that the kernels $k_X$, $k'_X$ and $\kappa$ are bounded by $B_k^2$, $B_{k'}^2 \geq 1$ and $B_{\kappa'}^2$, respectively, and the canonical feature map $\Phi_\kappa : v \in \mathcal{H}_{k'_X} \mapsto \kappa(v, \cdot) \in \mathcal{H}_\kappa$ of $\kappa$ is $L_\kappa$-Hölder of order $\alpha \in (0, 1]$ on the closed ball $B_{H_{k'_X}}(B_{k'})$. Then for any $r > 0$ and $\delta \in (0, 1)$, with probability at least $1 - \delta$, it holds that:*

$$\sup_{h \in \mathcal{B}_{\mathcal{H}_{\bar{k}}}(r)} |\hat{\xi}(h) - \xi(h)| \leq C\left( B_l\sqrt{-M^{-1}\log\delta} + rB_kL_l\left( B_{k'}L_\kappa\left( n^{-1}\log\left(M/\delta\right)\right)^{\alpha/2} + B_\kappa/\sqrt{M}\right)\right),$$

(2)

*where $C$ is a constant.*

*Theorem* 2 indicates that the error bound becomes larger in general if $(M, n)$ is replaced with $(1, Mn)$. Therefore, using domain-wise datasets is generally better than just pooling them into one mixed dataset, so the domain information plays a role (Wang et al., 2023). In other words, mixing the datasets would erase the information of domain identity, which could be otherwise valuable in algorithm design. Such theory is also intuitively valid, stimulating interests in studies.

Most efforts in DG consider learning domain-invariant representations, disentangling task-related information, or stabilizing the learning process with specific optimization strategies (Wang et al., 2023). Representation learning (Bengio et al., 2013) aims to learn domain-invariant representation with examples such as explicitly matching Maximum Mean Discrepancies (MMD) (Li et al., 2018b) of source representations or through adversarial network training of Domain-Adversarial Neural Networks (DANN) (Ganin et al., 2016) and its variant Class-conditional DANN (C-DANN) (Li et al., 2018c), MTL (Blanchard et al., 2021). Feature disentanglement considers dissecting samples to feature vectors that each dimension would reflect different factors of variation, with SagNets (Nam et al., 2019) being one example. Specific learning strategies for DG have also been explored, with notable ones such as MLDG (Li et al., 2018a), Group Distributionally Robust Optimization (GroupDRO) (Sagawa et al., 2020), Variance of Risk Extrapolation (VREx) (Krueger et al., 2021), Representation Self-Challenging (RSC) (Huang et al., 2020), Fish (Shi et al., 2022), and its variant Fishr (Rame et al., 2022). However, extensive experiments on DomainBed (Gulrajani & Lopez-Paz, 2021) results suggest that when carefully implemented and tuned, ERM outperforms most state-of-the-art approaches in terms of average performance, with CORAL (Sun et al., 2016) being the exception that has stable performance improvements.

Aside from the above genres, ensemble learning-based DG approaches in recent works started to shine due to better performance. One side of the work focuses on better optimization procedures through ensemble. Without resorting to the test domain, SWAD (Cha et al., 2021) and EoA (Arpit et al., 2022) use model ensemble and weight ensemble directly to improve the generalization. The other side of the work focuses on directly utilizing pre-trained models. SIMPLE (Li et al., 2023c)

adopts a large number of pre-trained models and aggregates without fine-tuning, pointing out that none of the pre-trained models can dominate in all unseen distributions.

### A.1.2    TEST-TIME TRANSFERABILITY ESTIMATION

To utilize the domain identity information, however, it does not have to be in the paradigm of single-model DG aiming for generalization. *Theorem* 2 emphasizes the importance of integrating domain identity information considering arbitrary test domains, but in reality, the task at hand is what matters, i.e., to achieve good performance on the specific test domain. When there are multiple source domain classifiers of $h$ available, minimizing the risks of each single predictor $h$ is another possible path.

One general idea of using test-time ensemble learning in DG is to utilize the relationship, or extent of transferability (Jiang et al., 2022), between the unseen domains and source domains. In deep learning, transferability refers to the ability of deep neural networks to learn from some source tasks and then adapt the gained knowledge to improve learning in related target tasks (Bengio, 2012).

Specifically, an extra domain classifier could be trained to predict the probability of each source domain that a test sample belongs to and aggregate domain-specific models' predictions by weights accordingly (Mancini et al., 2018). DRM (Zhang et al., 2023b) proposes to use prediction entropy as such measure of estimation. It uses a framework that trains only domain-specific classifier heads while letting them share the same feature extractor as also in Zhou et al. (2021). In addition, domain-specific batch normalization layers could also be used in the weighting for aggregation in inference (Segu et al., 2023).

Considering further real-time or online adaptation during the test phase, test-time adaptation (Liang et al., 2023) approaches could also be utilized in DG setting based on the specific application requirements. Such approaches usually need to further adapt the model parameters or introduce extra structures into the framework, and are not the focus of this work.

### A.1.3    UNSUPERVISED ENSEMBLE LEARNING

Ensemble learning primarily refers to supervised ensemble approaches, with bagging (Breiman, 1996) and boosting (Schapire, 1999) being the two most famous kinds. In contrast, unsupervised ensemble learning has received much less attention while being equally, if not more, important. Given the broad range of real-world applications, careful performance estimation of models instead of direct deployment is crucial for reliable machine learning. Deep learning-based approaches have been proposed under specific settings for unsupervised performance evaluation (Jiang et al., 2021; Chen et al., 2021; Baek et al., 2022). Ensemble learning further considers using such estimation for better aggregation from multiple learners. Unsupervised ensemble learning aims to measure observer errors on test data without resorting to any annotations.

The first work in unsupervised ensemble learning considers applying maximum likelihood estimation on the performance of multiple predictors on unlabeled test data using the expectation maximization (EM) algorithm (Dawid & Skene, 1979). An even more straightforward approach, the Spectral Meta-Learner (SML) (Parisi et al., 2014), constructs a weighted ensemble of predictors using only test set inferences of predictors with a few assumptions under binary cases. A fundamental assumption in their work was perfect diversity between the different predictors. Namely, the prediction errors were assumed statistically independent across predictors and test samples. Its main advantage is its almost analytic solution, saving the need for iterative optimization. Such approaches are extremely useful in fields such as gene analysis (Marbach et al., 2012; Ionita-Laza et al., 2016).

Later works provide more theoretical analysis and extensions to the SML, but are all either evaluated under synthetic or small datasets (Jaffe et al., 2015; 2016; Shaham et al., 2016; Wu et al., 2016; Li et al., 2023b), or have inherent limitations of usage, e.g., iterative training (Zhang et al., 2014; Traganitis et al., 2018), extra trainable parameters (Shaham et al., 2016), ranking approximation (Ahsen et al., 2019). There still lacks a theoretical and empirical study of SML under contemporary deep learning background, which could appropriately handle SML's expectation of an infinitely large unlabeled test set, assumption of perfect conditional independence between errors of predictors, and limitation of binary classification setting.

### A.1.4 MIXTURE-OF-EXPERTS AND META-LEARNING

Mixture-of-Experts (MoE) models, often incorporating meta-learning optimization strategies, have been explored in the DG context. MoE models implicitly learn weights for each respective expert through trainable model parameters, distinguishing them from the explicit weight calculation in SML.

General MoE approaches typically do not employ domain-specific experts. Generalizable Mixture-of-Experts (GMoE) (Li et al., 2023a) introduces sparse MoEs with a cosine router, emphasizing transformer architecture. Other MoE designs specifically incorporate domain-specific experts. For instance, in offline unsupervised domain adaptation, Guo et al. (2018) proposes an MoE that learns a point-to-set Mahalanobis distance metric to weigh the experts for different target examples, introducing meta-learning within the sources to address the unlabeled target domain. Given the inaccessibility of the target domain in DG, subsequent works often resort to the meta-learning paradigm, using support set and query set splits within the source domains to meta-learn the proper assignment of aggregation weights. Mixture of Domain-specific Experts (MoDE) (Kim et al., 2022), Relevance-aware MoE (RaMoE) (Dai et al., 2021), and Meta-Distillation of MoE (Meta-DMoE) (Zhong et al., 2022) fall under this concept. The choice of distance metrics may include L2-distance, cosine distance, or Mahalanobis distance, depending on the design. Typically, a small amount from the target domain serves as the support set to further optimize such models, as observed in Meta-DMoE or ARM.

Notably, MoE approaches focus on learning the metric function or ensemble weights of aggregation in the feature space during the training phase, while SML concentrates on analyzing cross-sample correlations in prediction probabilities during the test phase.

## A.2 MORE ILLUSTRATIVE FIGURES

We show domain-specific ERM and combined DG ERM results in Figure 6 on the PACS dataset. Observe that the extent of transferability varies across domains. We also show feature distributions using domain-invariant representation learning DG approaches in Figure 7. Observe that features from the more distinct test domains still have a large gap to the source feature distributions.

## A.3 PRELIMINARIES OF SPECTRAL UNSUPERVISED ENSEMBLE LEARNING

Most of the preliminaries here follow SML (Parisi et al., 2014), which are also in Jaffe et al. (2015; 2016); Traganitis et al. (2018).

Consider binary classification tasks where the number of classes is $K = 2$. $\mathcal{D}_t = \{(\mathbf{x}_i)\}_{i=1}^{n_t}$ test samples are been evaluated using $M \geq 3$ predictors $\{f_j\}_{j=1}^{M}$ which provide either positive or negative responses $f_j(\mathbf{x}_i) \in \{-1, 1\}$. The goal is to utilize the predictions $\{f_j(\mathbf{x}_i)\}$ of the predictors and reliably rank their performance without resorting to the true labels. More preferably, an ensemble learner could be built using the predictors that is able to achieve better classification results than uniform averaging or majority voting.

The performance of predictors are denoted as follows: The sensitivity $\psi$ and specificity $\eta$ could be formally defined as:

$$\psi = P[f(\mathbf{x}) = Y | Y = 1], \tag{3}$$

and

$$\eta = P[f(\mathbf{x}) = Y | Y = -1]. \tag{4}$$

The balanced classification accuracy of each predictor $f$ is denoted as $\pi$, which in binary case is defined as the mean of its sensitivity $\psi$ and specificity $\eta$:

$$\pi = \frac{\psi + \eta}{2}. \tag{5}$$

Spectral approach considers to rank the classifiers and builds a weighted ensemble or a meta-learner without labeled data. Let $\mathcal{Q}$ be the $M \times M$ population covariance matrix of the $M$ predictors, whose entries $q_{ij}$ are:

$$q_{ij} = \mathbb{E}[(f_i(\mathbf{x}) - \mu_i)(f_j(\mathbf{x}) - \mu_j)], \tag{6}$$

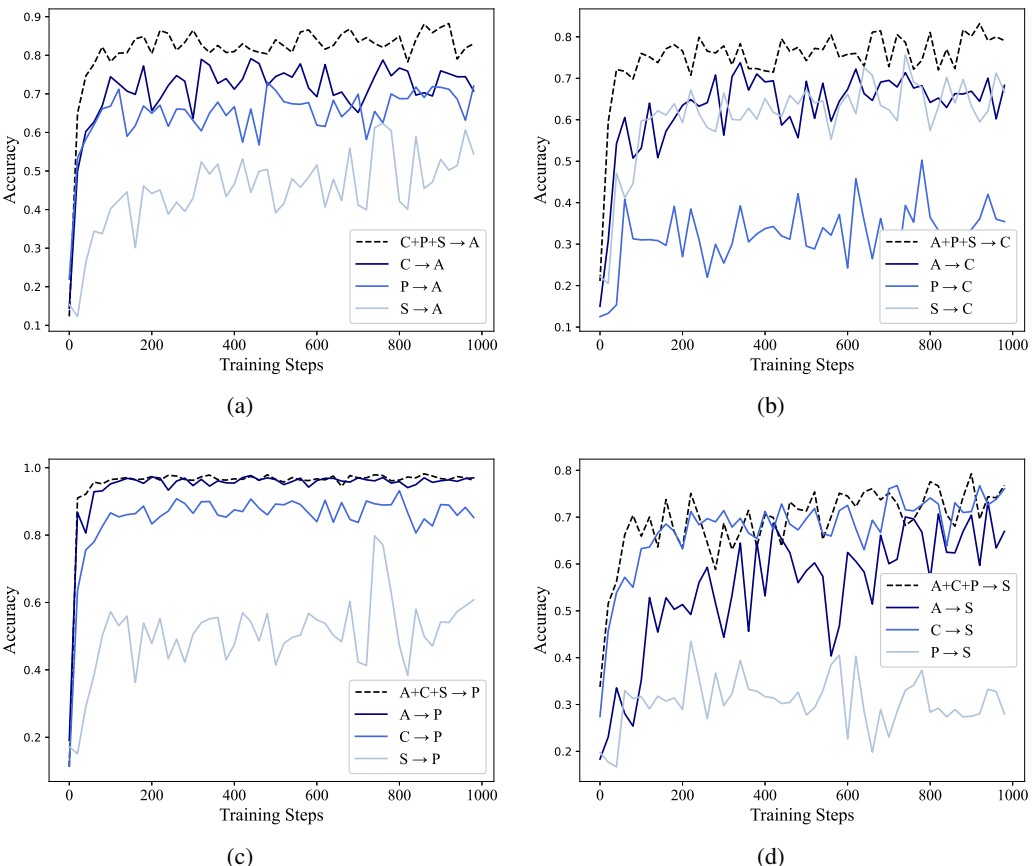

Figure 6: Domain-specific ERM and combined ERM results on test domain of (1) Art; (2) Cartoon; (3) Photo; and (4) Sketch.

where $\mathbb{E}$ denotes expectation with respect to the density $p(x, y)$ and $\mu_i = \mathbb{E}[(f_i(\mathbf{x})]$.

Under the following assumptions, *Lemma* 3 and *Lemma* 4 can be proved:

1. Test data $\mathcal{D}_t = \{(\mathbf{x}_i)\}_{i=1}^{n_t}$ satisfy the i.i.d. assumption.
2. Conditional independence between predictors $\{f_j\}_{j=1}^{M}$.
3. Most predictors have performance better than random.

**Lemma 3.** *(Parisi et al., 2014) The entries $q_{ij}$ of $\mathcal{Q}$ are equal to*

$$q_{ij} = \begin{cases} 1 - \mu_i^2 & i = j \\ (2\pi_i - 1)(2\pi_j - 1)(1 - b^2) & otherwise \end{cases} \tag{7}$$

*where $b \in (-1, 1)$ is the class imbalance:*

$$b = P[Y = 1] - P[Y = -1]. \tag{8}$$

The off-diagonal entries are identical to those of a rank-one matrix $R = \lambda \boldsymbol{v}\boldsymbol{v}^\top$ with unit-norm leading eigenvector $\boldsymbol{v}$ and eigenvalue $\lambda$ that

$$\lambda = (1 - b^2) \cdot \sum_{i=1}^{M} (2\pi_i - 1)^2, \tag{9}$$

More importantly,

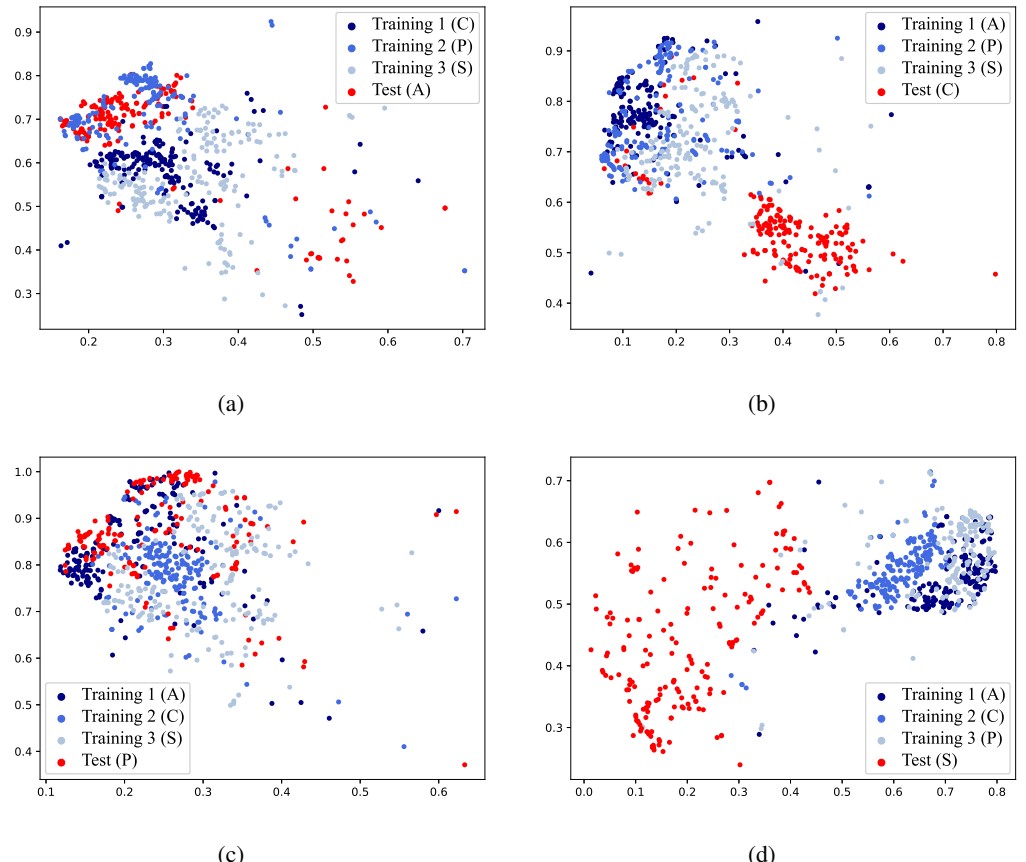

Figure 7: t-SNE visualization of samples from the Dog class with DANN DG on test domain of (1) Art; (2) Cartoon; (3) Photo; and (4) Sketch.

**Lemma 4.** *(Parisi et al., 2014) the entries of $v$ are proportional to the balanced accuracies of the $M$ predictors,*

$$v_i = \sqrt{1 - b^2}(2\pi_i - 1). \tag{10}$$

An ensemble can be constructed using the entries of $v$ for label prediction:

$$\hat{y}^{\text{SML}} = \text{sign}(\sum_{i=1}^{M} f_i(\mathbf{x}) \cdot \hat{v}_i), \tag{11}$$

where $\hat{v}_i$ is the estimated value of $v_i$.

There can be various estimation methods, i.e., (weighted) linear system, semi-definite programming, direct eigendecomposition, as discussed in Parisi et al. (2014). The SML ensemble is expected to be more accurate than majority voting in binary cases.

### A.4 PROOF OF *Lemma* 1

For multi-class case, i.e., there are $K > 2$ classes. Instead of either positive or negative responses $\{f_j(x_i)\} \in \{-1, 1\}$ in the binary case, predictors now return the predicted class id $\{f_j(x_i)\} \in \{1, 2, ..., K\}$. Each predictor is characterized by a $K \times K$ confusion matrix $\psi^i$ where

$$\psi_{kk'} = P(f(\mathbf{x}) = k | Y = k'), \tag{12}$$

Consider a split of the group $\mathcal{Y} = \{1, 2, ..., K\}$ into two non-empty disjoint subsets, $\mathcal{Y} = \mathcal{A} \cup (\mathcal{Y} \setminus \mathcal{A})$ and the binary predictors $\{f_i^{\mathcal{A}}\}_{i=1}^M$:

$$f_i^{\mathcal{A}}(\mathbf{x}) = \begin{cases} 1 & f_i(x) \in \mathcal{A} \\ -1 & f_i(x) \notin \mathcal{A} \end{cases} \tag{13}$$

Using the previous formulas along with *Lemma* 4 and *Lemma* 3 in the binary case, the probability of group $\mathcal{A}$

$$p^{\mathcal{A}} = P(\mathcal{Y} \in \mathcal{A}) = \sum_{k \in \mathcal{A}} p_k \tag{14}$$

and the sensitivity of each predictor $f_i^{\mathcal{A}}$ can be estimated. Hence, by considering all one-vs-all splits of classes, all class probabilities and all diagonal entries $\psi_{kk'}$ can be estimated.

We prove *Lemma* 1 of multi-class SML under the following assumptions:

1. Test data $\mathcal{D}_t = \{(\mathbf{x}_i)\}_{i=1}^{n_t}$ satisfy the i.i.d. assumption.

2. Conditional independence between predictors $\{f_j\}_{j=1}^M$.

3. Most predictors have performance better than random.

4. Test data $\mathcal{D}_t = \{(\mathbf{x}_i)\}_{i=1}^{n_t}$ are class-balanced.

Note that we have an extra fourth assumption compared to the binary case.

In multi-class SML, we consider all one-vs-rest splits of classes, i.e., for each target class $k$, $\mathcal{A}_k \cup (\mathcal{Y} \setminus \mathcal{A}_k)$, where $\mathcal{A}_k = \{k\}$ and $\mathcal{Y} \setminus \mathcal{A}_k = \{1, 2, ..., k-1, k+1, ..., K\}$. The sensitivity of the predictor is thus

$$\psi_{(k)} = P[f(\mathbf{x}) = k | Y = k], \tag{15}$$

and

$$\eta_{(k)} = P[f(\mathbf{x}) \neq k | Y \neq k]. \tag{16}$$

Obviously, their average is no longer the balanced classification accuracy over all the $K$ classes.

$$\pi_{(k)} = \frac{\psi_{(k)} + \eta_{(k)}}{2} = \frac{\sum_{k=1}^K (P[f(\mathbf{x}) = k | Y = k] + P[f(\mathbf{x}) \neq k | Y \neq k])}{2K} \tag{17}$$

$$\neq \frac{\sum_{k=1}^K P[f(\mathbf{x}) = k | Y = k]}{K}, \tag{18}$$

The extra term introduces confusion in the estimation. Regardless, both *Lemma* 3 and *Lemma* 4 still hold for all $\mathcal{A}_k \cup (\mathcal{Y} \setminus \mathcal{A}_k)$ splits in their binary task, i.e., values of each $\boldsymbol{v}_{(k)}$ are proportional to $\pi_{(k)}$:

$$v_{i(k)} = \sqrt{1 - b_{(k)}^2}(2\pi_{i(k)} - 1). \tag{19}$$

Note that the proportion $\sqrt{1 - b_{(k)}^2}$ is equivalent across all $K$ classes under assumption 4 as class imbalance $b$ is independent of $k$:

$$b_{(k)} = P[Y = k] - [Y \neq k] = \frac{1}{K} - \frac{K-1}{K} = \frac{2-K}{K}, \tag{20}$$

and $b_{(k)}$ will be simply denoted as $b$ later on.

Therefore, the weights $\{\boldsymbol{v}_{(k)}\}_{k=1}^K$ across $K$ splits of $\mathcal{A}_k \cup (\mathcal{Y} \setminus \mathcal{A}_k)$ could be averaged, and the values of the average weight vector are:

$$\bar{v}_j = \frac{1}{K} \sum_{k=1}^K v_{j(k)} = \frac{\sqrt{1 - b^2}}{K} \sum_{k=1}^K (2\pi_{j(k)} - 1), \tag{21}$$

where the performance metric, instead of the balanced classification accuracy metric in binary setting, is denoted as the $\hat{\pi}$-performance:

$$\hat{\pi}_j = \frac{1}{K} \sum_{k=1}^{K} \pi_{j(k)} = \frac{1}{K} \sum_{k=1}^{K} \frac{P[f_j(\mathbf{x}) = k|Y = k] + P[f_j(\mathbf{x}) \neq k|Y \neq k]}{2}. \tag{22}$$

Therefore, we build the multi-class SML using one-vs-rest splitting, and the final prediction label is:

$$\hat{y}_i^{\text{Multi-Class SML}} = \arg\max_k \left( \sum_{j=1}^{M} \delta_k \left( f_j \left( \mathbf{x}_i \right) \right) \cdot \bar{v}_j \right), \tag{23}$$

where $\hat{v}_j$ is the estimation of $\bar{v}_j$ using similar approach for binary SML.

## A.5 DATASET INFORMATION

Dataset statistics are summarized in Table 3 and sample images are shown in Figure 8.

Table 3: Statistics of the used Datasets.

| Dataset | # Domains | # Classes | # Total Samples |
|---|---|---|---|
| VLCS | 4 | 5 | 10,729 |
| PACS | 4 | 7 | 9,991 |
| OfficeHome | 4 | 65 | 15,588 |
| TerraIncognita | 4 | 10 | 24,330 |
| DomainNet | 6 | 345 | 586,575 |

## A.6 ANALYSIS OF ONLINE INFERENCE

The proposed multi-class SML in SMEE has a few assumptions and utilizes model predictions test domain. It is interesting to investigate how much degradation to the performance it would cause from switching it from offline to online incremental setting, i.e., test samples arrive in a stream one by one, instead of being completely available for formula 1 calculation. The results are shown in Figure 9, where a 'spike' means that the online inference result differs from the offline inference result on that sample. The runs were randomly selected using the first domain in each dataset as the test domain and five meta-selected ResNet-50 models per source in SMEE ensemble.

Observe that there is almost no difference in prediction when switching the algorithm offline to online for datasets PACS and VLCS, possibly due to their relatively high transfer performance, smaller test sets, and fewer number of classes. We have a higher amount of prediction difference on TerraIncognita with lower transfer performance and OfficeHome with a relatively higher number of classes. However, they are still well below 0.5% of predictions for TerraIncognita and 2% for OfficeHome. More importantly, such differences in predictions mostly happen at the beginning of testing, where the online version has to resort to uniform-weights averaging due to the inability to calculate the covariance matrix when $i \leq M$. It is intuitive as we have less test domain information at the beginning of the test phase, and the covariance matrix estimation may have a higher error. Do note that the difference does not necessarily lead to wrong classification results. It also suggests that after accumulating a certain amount of test domain samples, enough information from the test domain could be accumulated to accurately determine the performance ranking of predictors.

## A.7 BINARY TREE APPROXIMATION FOR LARGE NUMBER OF CLASSES

For datasets with many classes and test samples, the loop through all one-vs-all splits would be rather time-consuming. Binary tree splits of classes approximation could be used instead of the entire one-vs-rest splits in SMEE calculation. Specifically, for a loop number $N \in \{1, 2, ..., \lceil \log_2 K \rceil\}$ splits, the classes are first split into $min(2^N, K)$ leaf nodes based on a binary tree structure. We

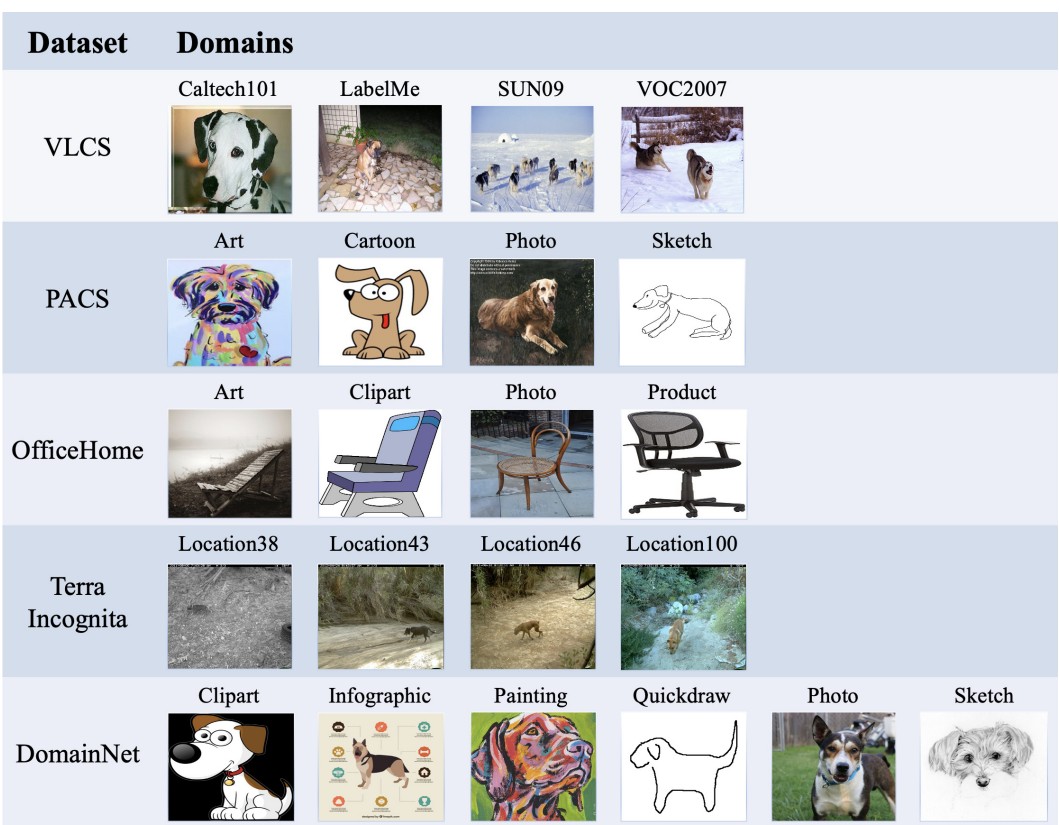

Figure 8: Sample Images from the used datasets.

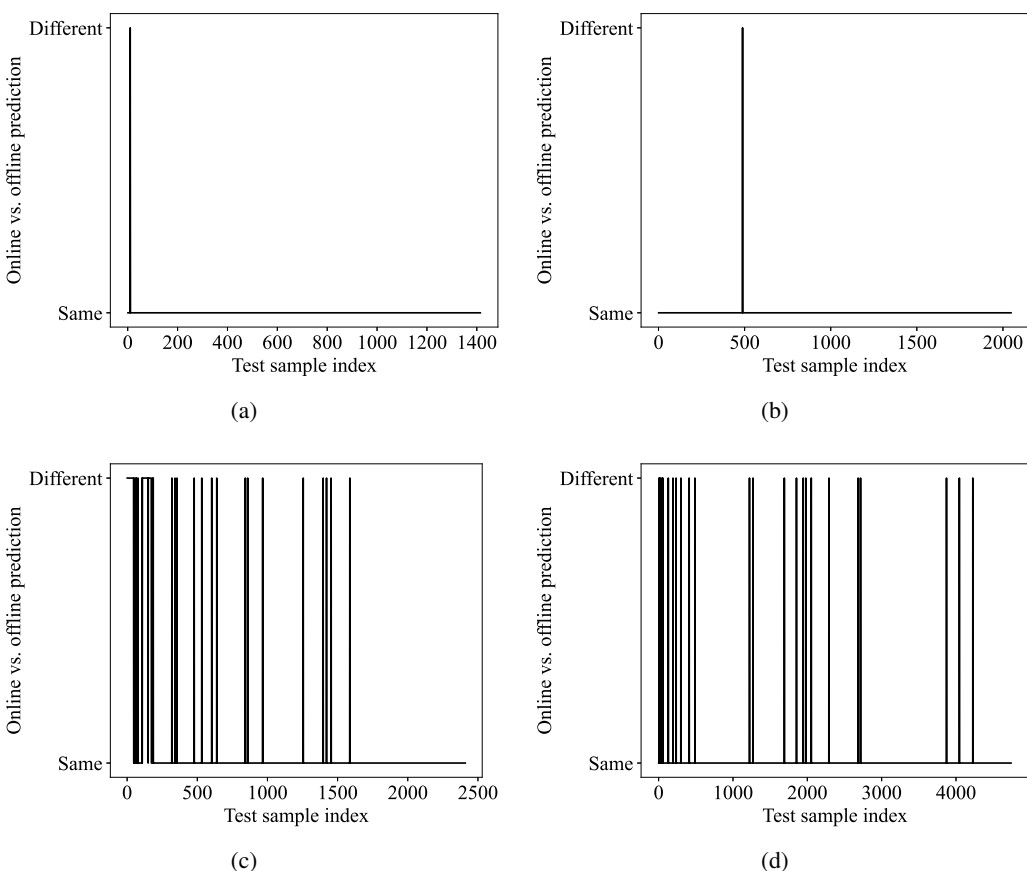

Figure 9: Comparison of offline and online inference results of multi-class SML in SMEE on the four datasets (a) PACS; (b) VLCS; (c) OfficeHome; and (d) TerraIncognita. A 'spike' means that the offline and online prediction results differ on that test sample.

randomly choose half of the leaf nodes where none of the chosen nodes have the same parent node, and combine them as the positive class, where the remaining ones are treated as the negative class in the split. In such a way, we reduce the number of loops from $K$ to $\lceil \log_2 K \rceil$.

The performance comparison of using nine splits of binary tree approximation splits of classes instead of the entire 345 one-vs-rest splits is shown in Figure 10. Observe that using the binary tree approximation results in a performance drop, but multi-class SML still performs better than uniform-weights averaging. Since we aim for real-time online inference in this work, which presumably should immediately return prediction results for test samples in a stream, such approximation could be used. If computational time is not a consideration, using the full splits would result in better performance.

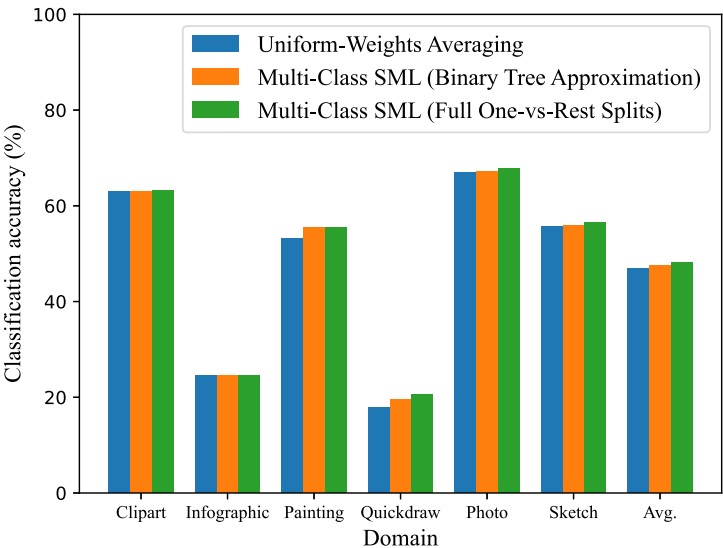

Figure 10: Comparison of one-vs-rest split and binary tree approximation split of classes in multi-class SML in SMEE on the DomainNet dataset for each domain. Five domain-specific experts are meta-selected from each source domain of ten independently initialized ResNet-50 models, and then utilized on the test domain online for aggregation.

## A.8 COMPUTATIONAL COST

The maximum computational cost on the used datasets using an Intel(R) Xeon(R) Platinum 8176 CPU @ 2.10GHz is shown in Table 4. Except for DomainNet, which is a substantially larger dataset, multi-class SML can be executed with minimal computational effort in milliseconds.

It is important to note that the computation time of multi-class SML scales proportionally with the number of classes. In real-world scenarios, especially for large test set of datasets like DomainNet, recalculating the covariance matrix for each new incoming test sample becomes impractical, even with an approximation strategy. We discuss how to address this challenge in real-world scenarios below. In Section A.6, we analyzed the differences between offline and complete online approaches, revealing that the disparity between them is relatively small. Importantly, this disparity primarily occurs at the beginning of testing. Additionally, the objective of multi-class SML, which involves calculating weights for each expert in the ensemble, does not require uniform updates for each individual test sample, both at the beginning and towards the end.

To minimize computation costs for large test set with numerous classes, we propose a straightforward strategy. Initially, during the test phase, the matrix size remains small, facilitating efficient updates of ensemble weights for every test sample. As demonstrated in the table, this process takes only milliseconds. Once a sufficient number of test sample predictions have been accumulated, updating the weights for each new sample introduces minor changes to the ensemble weights. Consequently, the update can now be performed after a batch of test queries. This update can be scheduled at

intervals, presumably during periods when the system is idle and has abundant computation resources. The exact point of this transition depends on the computational resources available and the actual dataset in practice. To gauge when this transition point should occur, we refer to Figure 9 where we compare offline and online prediction disparities.

Table 4: The computation time of multi-class SML on respective datasets at its maximum using the entire test set. The first domain in each dataset was employed for illustration.

| Dataset | SVD matrix maximum size (# experts, # samples, # classes) | computation time (seconds) |
|---|---|---|
| **VLCS** | (15, 1415, 5) | 0.008 |
| **PACS** | (15, 2048, 7) | 0.019 |
| **OfficeHome** | (15, 2427, 65) | 0.535 |
| **TerraIncognita** | (15, 4741, 10) | 0.109 |
| **DomainNet** | (25, 48129, 345) | 61.423 |

## A.9 EXTENDED RESULTS

This subsection shows the results for a complete list of baselines we could find in Table 5. The extra baselines include MMD (Li et al., 2018b), GroupDRO (Sagawa et al., 2020), Invariant Risk Minimization (IRM) (Arjovsky et al., 2019), Adaptive Risk Minimization (ARM) (Zhang et al., 2021), VREx (Krueger et al., 2021), DANN (Ganin et al., 2016), C-DANN (Li et al., 2018c), and RSC (Huang et al., 2020).

Results of multi-experts test-time transferability estimation and ensemble for each test domain are shown in Tables 6-10 for the five datasets respectively.

Table 5: Average classification accuracies on five datasets with more baselines. Baseline results are from either original publications or reproduced by DomainBed (Gulrajani & Lopez-Paz, 2021), under the same setup. The bracket in model zoo ensemble indicates the number of models used in the ensemble. The best are marked in bold, and the second best by an underline.

| Algorithm | VLCS | PACS | OfficeH. | TerraInc. | DomainN. | Avg. |
|---|---|---|---|---|---|---|
| *Non-ensemble Algorithms* | | | | | | |
| MMD (Li et al., 2018b) | $77.5_{\pm0.9}$ | $84.6_{\pm0.5}$ | $66.3_{\pm0.1}$ | $42.2_{\pm1.6}$ | $23.4_{\pm9.5}$ | 59.5 |
| GroupDRO (Sagawa et al., 2020) | $76.7_{\pm0.6}$ | $84.4_{\pm0.8}$ | $66.0_{\pm0.7}$ | $43.2_{\pm1.1}$ | $33.3_{\pm0.2}$ | 61.4 |
| IRM (Arjovsky et al., 2019) | $78.5_{\pm0.5}$ | $83.5_{\pm0.8}$ | $64.3_{\pm2.2}$ | $47.6_{\pm0.8}$ | $33.9_{\pm2.8}$ | 61.6 |
| ARM (Zhang et al., 2021) | $77.6_{\pm0.3}$ | $85.1_{\pm0.4}$ | $64.8_{\pm0.3}$ | $45.5_{\pm0.3}$ | $35.5_{\pm0.2}$ | 62.4 |
| VREx (Krueger et al., 2021) | $78.3_{\pm0.2}$ | $84.9_{\pm0.6}$ | $66.4_{\pm0.6}$ | $46.4_{\pm0.6}$ | $33.6_{\pm2.9}$ | 62.5 |
| CDANN (Li et al., 2018c) | $77.5_{\pm0.1}$ | $82.6_{\pm0.9}$ | $65.8_{\pm1.3}$ | $45.8_{\pm1.6}$ | $38.3_{\pm0.3}$ | 62.6 |
| DANN (Ganin et al., 2016) | $78.6_{\pm0.4}$ | $83.6_{\pm0.4}$ | $65.9_{\pm0.6}$ | $46.7_{\pm0.5}$ | $38.3_{\pm0.1}$ | 63.2 |
| RSC (Huang et al., 2020) | $77.1_{\pm0.5}$ | $85.2_{\pm0.9}$ | $65.5_{\pm0.9}$ | $46.6_{\pm1.0}$ | $38.9_{\pm0.5}$ | 63.2 |
| ERM (Gulrajani & Lopez-Paz, 2021) | $77.5_{\pm0.4}$ | $85.5_{\pm0.2}$ | $66.5_{\pm0.3}$ | $46.1_{\pm1.8}$ | $40.9_{\pm0.1}$ | 63.3 |
| MTL (Blanchard et al., 2021) | $77.2_{\pm0.4}$ | $84.6_{\pm0.5}$ | $66.4_{\pm0.5}$ | $45.6_{\pm1.2}$ | $40.6_{\pm0.1}$ | 63.4 |
| Mixup (Xu et al., 2020) | $77.4_{\pm0.6}$ | $84.6_{\pm0.6}$ | $68.1_{\pm0.3}$ | $47.9_{\pm0.8}$ | $39.2_{\pm0.1}$ | 63.4 |
| MLDG (Li et al., 2018a) | $77.2_{\pm0.4}$ | $84.9_{\pm1.0}$ | $66.8_{\pm0.6}$ | $47.7_{\pm0.9}$ | $41.2_{\pm0.1}$ | 63.6 |
| Fish (Shi et al., 2022) | $77.8_{\pm0.3}$ | $85.5_{\pm0.3}$ | $68.6_{\pm0.4}$ | $45.1_{\pm1.3}$ | $42.7_{\pm0.2}$ | 63.9 |
| CORAL (Sun et al., 2016) | $78.8_{\pm0.6}$ | $86.2_{\pm0.3}$ | $68.7_{\pm0.3}$ | $47.6_{\pm1.0}$ | $41.5_{\pm0.1}$ | 64.6 |
| mDSDI (Bui et al., 2021) | $79.0_{\pm0.3}$ | $86.2_{\pm0.2}$ | $69.2_{\pm0.4}$ | $48.1_{\pm1.4}$ | $42.8_{\pm0.1}$ | 64.6 |
| SagNet (Nam et al., 2019) | $77.8_{\pm0.5}$ | $86.3_{\pm0.2}$ | $68.1_{\pm0.1}$ | $48.6_{\pm1.0}$ | $40.3_{\pm0.1}$ | 64.7 |
| Fishr (Rame et al., 2022) | $78.2_{\pm0.2}$ | $86.9_{\pm0.2}$ | $68.2_{\pm0.2}$ | $53.6_{\pm0.4}$ | $41.8_{\pm0.2}$ | 65.7 |
| MIRO (Cha et al., 2022) | $79.0_{\pm0.0}$ | $85.4_{\pm0.4}$ | $70.5_{\pm0.4}$ | $50.4_{\pm1.1}$ | $44.3_{\pm0.2}$ | 65.9 |
| AdaNPC (Zhang et al., 2023c) | $79.5_{\pm2.4}$ | $88.8_{\pm0.1}$ | / | $53.9_{\pm0.3}$ | $42.9_{\pm0.5}$ | / |
| *Ensemble Algorithms* | | | | | | |
| Meta-DMoE (Zhong et al., 2022) | / | 86.9 | / | / | 44.2 | / |
| DRM (Zhang et al., 2023b) | $\underline{80.5}_{\pm0.3}$ | $88.5_{\pm1.2}$ | / | / | $42.4_{\pm0.1}$ | / |
| SWAD (Cha et al., 2021) | $79.1_{\pm0.1}$ | $88.1_{\pm0.1}$ | $70.6_{\pm0.2}$ | $50.0_{\pm0.3}$ | $46.5_{\pm0.1}$ | 66.9 |
| GMoE (Li et al., 2023a) | $80.2_{\pm0.2}$ | $88.1_{\pm0.1}$ | $\mathbf{74.2}_{\pm0.4}$ | $48.5_{\pm0.1}$ | $\mathbf{48.7}_{\pm0.2}$ | 67.9 |
| EoA (Arpit et al., 2022) | 79.1 | $\underline{88.6}$ | 72.5 | 52.3 | 47.4 | 68.0 |
| MIRO + SWAD (Cha et al., 2022) | $79.6_{\pm0.2}$ | $88.4_{\pm0.1}$ | $72.4_{\pm0.1}$ | $\mathbf{52.9}_{\pm0.2}$ | $47.0_{\pm0.0}$ | $\underline{68.1}$ |
| SMEE (ours) | $\mathbf{82.1}_{\pm0.4}$ | $\mathbf{90.4}_{\pm0.4}$ | $\underline{73.2}_{\pm0.6}$ | $\underline{52.8}_{\pm2.1}$ | $\underline{48.1}_{\pm0.6}$ | $\mathbf{69.3}$ |
| *Model Zoo Ensemble* | | | | | | |
| SIMPLE (15) (Li et al., 2023c) | $79.8_{\pm0.1}$ | $84.1_{\pm0.5}$ | $79.9_{\pm0.1}$ | $56.8_{\pm0.2}$ | $46.3_{\pm0.4}$ | 69.4 |
| SIMPLE (224) (Li et al., 2023c) | $79.9_{\pm0.5}$ | $88.6_{\pm0.4}$ | $\mathbf{84.6}_{\pm0.5}$ | $\mathbf{57.6}_{\pm0.8}$ | $\mathbf{49.2}_{\pm1.1}$ | $\mathbf{72.0}$ |
| SMEE (15-25) (ours) | $\mathbf{84.2}_{\pm0.6}$ | $\mathbf{91.6}_{\pm0.6}$ | $78.4_{\pm0.3}$ | $55.4_{\pm1.6}$ | $48.7_{\pm0.6}$ | 71.7 |

Table 6: Average classification accuracies on VLCS for each domain as the test domain. Five domain-specific experts are meta-selected from each source domain of ten independently initialized ResNet-50 models, and then utilized on the test domain online for aggregation.

| Algorithm | Caltech101 | LabelMe | SUN09 | VOC2007 | Avg. |
|---|---|---|---|---|---|
| Single | 89.3 | 59.5 | 64.3 | 66.9 | 70.0 |
| Voting | 96.6 | 62.0 | 80.2 | 81.6 | 80.1 |
| Averaging | 96.6 | 62.6 | 80.2 | 81.7 | 80.3 |
| CSM | **98.9** | 66.2 | 81.4 | 80.3 | 81.7 |
| L2SM | 98.6 | 66.4 | 80.6 | 80.0 | 81.4 |
| PEM | 95.6 | 64.9 | 78.1 | 77.2 | 78.9 |
| PUM | 96.7 | 63.3 | 80.5 | 81.2 | 80.4 |
| SML-soft | **98.9** | **66.8** | 81.5 | 82.7 | **82.5** |
| SMEE (ours) | **98.9** | 64.5 | **81.9** | **83.2** | 82.1 |

Table 7: Average classification accuracies on PACS for each domain as the test domain. Five domain-specific experts are meta-selected from each source domain of ten independently initialized ResNet-50 models, and then utilized on the test domain online for aggregation.

| Algorithm | Art | Cartoon | Photo | Sketch | Avg. |
|---|---|---|---|---|---|
| Single | 74.2 | 66.8 | 84.7 | 58.3 | 71.0 |
| Voting | 89.9 | 84.2 | 98.0 | 79.3 | 87.8 |
| Averaging | 90.5 | 85.0 | **98.7** | 80.1 | 88.6 |
| CSM | 89.9 | 84.8 | 98.2 | 80.1 | 88.3 |
| L2SM | 89.8 | 87.2 | 98.3 | 81.8 | 89.3 |
| PEM | 90.2 | 85.5 | 98.5 | 80.5 | 88.7 |
| PUM | 91.8 | 82.2 | 98.6 | 79.6 | 88.1 |
| SML-soft | **92.8** | 69.9 | 98.2 | 79.0 | 85.0 |
| SMEE (ours) | 91.4 | **88.5** | 98.5 | **83.1** | **90.4** |

Table 8: Average classification accuracies on OfficeHome for each domain as the test domain. Five domain-specific experts are meta-selected from each source domain of ten independently initialized ResNet-50 models, and then utilized on the test domain online for aggregation.

| Algorithm | Art | Clipart | Product | Photo | Avg. |
|---|---|---|---|---|---|
| Single | 50.1 | 44.8 | 61.6 | 63.3 | 54.9 |
| Voting | 69.5 | 59.1 | 79.0 | 81.1 | 72.2 |
| Averaging | 70.3 | 59.6 | 79.4 | **82.1** | 72.9 |
| CSM | **70.6** | **60.1** | 78.0 | 81.4 | 72.5 |
| L2SM | 70.4 | **60.1** | 78.0 | 81.6 | 72.5 |
| PEM | 66.7 | 57.3 | 76.5 | 79.0 | 69.9 |
| PUM | 70.0 | **60.1** | 79.4 | 81.8 | 72.8 |
| SML-soft | 69.8 | 59.5 | **80.3** | 81.9 | 72.9 |
| SMEE (ours) | **70.6** | 60.0 | 80.2 | 82.0 | **73.2** |

Table 9: Average classification accuracies on TerraIncognita for each domain as the test domain. Five domain-specific experts are meta-selected from each source domain of ten independently initialized ResNet-50 models, and then utilized on the test domain online for aggregation.

| Algorithm | L100 | L38 | L43 | L46 | Avg. |
|---|---|---|---|---|---|
| Single | 54.2 | 38.8 | 35.0 | 33.3 | 40.6 |
| Voting | 69.1 | 45.6 | 39.6 | 40.6 | 48.7 |
| Averaging | 68.8 | 47.0 | 43.8 | 41.0 | 50.2 |
| CSM | 67.3 | 47.0 | 50.6 | 38.9 | 51.0 |
| L2SM | 68.1 | 47.2 | 48.9 | 39.5 | 50.9 |
| PEM | 66.7 | 48.8 | 43.1 | 41.4 | 50.0 |
| PUM | **69.3** | 47.2 | 44.4 | 41.0 | 50.5 |
| SML-soft | 67.4 | **50.0** | **52.3** | 37.9 | 51.9 |
| SMEE (ours) | 68.1 | 46.8 | 50.2 | **46.1** | **52.8** |

Table 10: Average classification accuracies on DomainNet for each domain as the test domain. Five domain-specific experts are meta-selected from each source domain of ten independently initialized ResNet-50 models, and then utilized on the test domain online for aggregation.

| Algorithm | Clipart | Infographic | Painting | Quickdraw | Photo | Sketch | Avg. |
|---|---|---|---|---|---|---|---|
| Single | 37.2 | 16.0 | 29.9 | 12.7 | 38.7 | 32.1 | 23.5 |
| Voting | 61.5 | 24.0 | 51.3 | 17.6 | 64.9 | 54.1 | 45.6 |
| Averaging | 62.9 | **24.6** | 53.2 | 17.9 | 66.9 | 55.7 | 46.9 |
| CSM | 63.2 | 22.1 | 53.2 | 17.3 | 66.9 | 55.7 | 46.4 |
| L2SM | 62.5 | 21.8 | 53.8 | 17.2 | 65.2 | 56.1 | 46.1 |
| PEM | 59.1 | 23.5 | 50.4 | 16.9 | 63.9 | 52.7 | 44.4 |
| PUM | 62.4 | 24.4 | 52.8 | 17.9 | 66.6 | 55.3 | 46.6 |
| SML-soft | 56.3 | 20.4 | 45.6 | 16.9 | 60.9 | 49.2 | 41.5 |
| SMEE (ours) | **63.3** | **24.6** | **55.4** | **20.6** | **67.7** | **56.6** | **48.1** |

