# OpenReview forum: "Generalization or Specificity? Spectral Meta Estimation and Ensemble (SMEE) with Domain-specific Experts"
_ICLR.cc/2024/Conference — Submitted to ICLR 2024_

### Official Review · Reviewer_6gJi · 2023-10-29

**Soundness:** 4 excellent
**Presentation:** 4 excellent
**Contribution:** 4 excellent
**Rating:** 8
**Confidence:** 4

**Summary:**

This paper points out limitations in existing multi-source domain generalization methods, which struggle to generalize by training a model with diverse source domains. The authors propose a new technique for effectively filtering and ensembling individual expert models. Specifically, they introduce an unsupervised spectral ensemble approach, expanding on the Spectral Meta-Learner by using a one-vs-rest paradigm for multi-class classification. This proposed method significantly outperforms traditional single model approaches and even surpasses self-ensembling and multiple model ensemble techniques.

**Strengths:**

- The paper boasts high-quality writing that is very readable. It provides detailed comparisons with existing methods. Especially commendable is the clarity and precision in potentially challenging statements.

- The proposed method is both intuitive and powerful, and the paper thoroughly describes the underlying assumptions and preliminaries supporting its feasibility.

- In a multi-source domain generalization setting, the proposed approach notably surpasses not only the performance of a single model but also that of self-ensembling and multi-model ensemble methods.

**Weaknesses:**

The paper should compare the computational overhead of the proposed spectral meta estimation with other ensemble methods. Even if a direct measurement of all test times isn't possible, including a discussion on computational complexity would be beneficial. Without such comparisons or analyses, it's hard to gauge the practicality of the proposed method.

**Questions:**

There is no question.

---

> ### Author Response · Authors · 2023-11-17
> **Rebuttal-Reviewer 6gJi-(1/2)**
>
> Thanks for your encouraging words and constructive comments! We greatly appreciate your time and efforts in reading the paper and raising concerns, and our responses to your comments are given below.
>
> 1. *The paper should compare the computational overhead of the proposed spectral meta estimation with other ensemble methods. Even if a direct measurement of all test times isn't possible, including a discussion on computational complexity would be beneficial. Without such comparisons or analyses, it's hard to gauge the practicality of the proposed method.*
>
> Response: Thank you for the insightful comment! The computational complexity in an online incremental setting is indeed crucial in determining the applicability of the approach to specific real-world scenarios. Reviewer ExFD also raised the same concern. Building on your feedback, we have expanded our analysis and discussion on the computation cost, incorporating these improvements into our revision.
>
> The maximum computation cost for each of the dataset used was recorded, on a Intel(R) Xeon(R) Platinum 8176 CPU @ 2.10GHz. The results are shown in Table 4 of the paper, and we also present them below for your easy reference:
>
> | Dataset    |  SVD matrix maximum size (# experts, # samples, # classes) | Computation time (seconds) |
> | -------- | ------- | ------- |
> | VLCS  | (15, 1415, 5) | 0.008   |
> | PACS | (15, 2048, 7) | 0.019    |
> | OfficeHome  | (15, 2427, 65) | 0.535   |
> | TerraIncognita  | (15, 4741, 10) | 0.109   |
> | DomainNet |  (25, 48129, 345) | 61.423   |
>
> Note that the table showed the maximum computation time. In online incremental setting, this is the time required for the very last test sample. Except for DomainNet, a substantially larger dataset, multi-class SML can be executed with minimal computational effort in milliseconds. As an extra note of information, the forward pass of ResNet-50 on a single test sample takes around 0.2 second for that same CPU.
>
> It is important to note that the computation time of multi-class SML scales proportionally with the number of classes. In real-world scenarios, especially for datasets like DomainNet, recalculating the covariance matrix for each new incoming test sample becomes impractical. We discuss how to address this challenge in real-world scenarios below.
>
> In Appendix Section A.6, we analyze the differences between offline and complete online versions of multi-class SML. The difference is whether we can use the full test set information, or only a portion of the test set until this specific test sample for the covariance matrix calculation. The results revealed that the disparity between them is relatively small (almost zero difference for small datasets and 0.5%-2% for larger datasets). Importantly, this disparity primarily occurs at the beginning of testing.
>
> To minimize computation costs for large test set with numerous classes, we propose a straightforward strategy. Initially, during the test phase, the matrix size remains small, facilitating efficient updates of ensemble weights for every test sample. As demonstrated in the table, this process takes only milliseconds. Once a sufficient number of test sample predictions have been accumulated, updating the weights for each new sample would only introduce minor changes to the ensemble weights. Consequently, the update can now be performed after a batch of test queries.
>
> **To summarize**, ideally, the update to ensemble weights should be done for every single test sample at the beginning of the test phase. When we have accumulated enough test sample predictions, this update can be scheduled at intervals, presumably during periods when the system is idle and has abundant computation resources. The exact point of this transition depends on the computational resources available and the actual dataset in practice. To gauge when this transition point should occur, we refer to Figure 9, where we compare offline and online prediction disparities.
>
> We further specifically discuss three possible deployment settings:
>
> 1. **Experts as APIs offered by companies:** These APIs return only prediction results. This is where multi-class SML shines, operating solely on prediction probabilities without requiring access to model parameters, as discussed in Section 3.4. The computation can be performed on the user's end, such as a smartphone, using the strategy discussed above. The primary computation overhead would likely be the communication time of the APIs in this case.
>
> 2. **Experts deployed on the user's end with multiple available CPUs:** In this scenario, parallel computing of the forward passes of respective experts is feasible. The computation overhead would depend on both the matrix operations of multi-class SML and the forward passes on the CPUs. For DRM or SIMPLE, the computation overhead would depend on both their ensemble strategy calculation and the forward passes on the CPUs. The actual time depends on the architectures of models.

---

> ### Author Response · Authors · 2023-11-17
> **Rebuttal-Reviewer 6gJi-(2/2)**
>
> (continued)
>
> 3. **Single CPU executing forward passes of multiple experts:** In cases where only a single CPU is available for executing the forward passes of multiple experts, the test-ensemble strategy operates slowly. Here, the computation overhead would be (#experts * forward_pass_time + ensemble_cost) for multi-class SML, DRM, and SIMPLE. The computation time of multi-class SML is not vital in this case.
>
> To summarize, the computation time really depends on the actual scenario in practice. Our approach could be applicable to multiple scenarios with specific forms of adaptations, with the strategy discussed above.
>
> We appreciate your careful read and thank you a lot for your time in enhancing the paper!

---

### Official Review · Reviewer_52us · 2023-10-29

**Soundness:** 2 fair
**Presentation:** 3 good
**Contribution:** 1 poor
**Rating:** 1
**Confidence:** 5

**Summary:**

In contrast with the conventional single-model domain generalization framework, the paper proposes an alternative perspective that the knowledge of the unseen target domain is transferred from multiple source domains. Specifically, the prediction of unseen target data is an aggregate of the predictions from each individual source domain model, and a spectral unsupervised ensembling method is used to determine the contribution of each source model to the target. To further facilitate the selection of source models, the paper proposes a meta performance estimation technique that aims to filter out the underperformed models within the ensemble. The proposed method's effectiveness is validated across multiple benchmark datasets.

**Strengths:**

* The paper extends the Spectral Meta-Learner (SML) unsupervised ensemble learning approach from binary to multi-class classification as a way to aggregate the knowledge from multi-source domain models to the unseen target domain.

**Weaknesses:**

* The research lacks novelty in its claim. The problem addressed in this paper, namely Test-time adaptation for Distribution shifts, was comprehensively discussed in [1]. Furthermore, the approach of aggregating knowledge from multiple source domains to an unseen target domain, often using techniques like Mixture-of-Experts or ensemble, is not a novel concept either [2, 3]

* Conceptually, the research presented in this paper bears a significant resemblance to [3]. Both works utilize a similar approach, pretraining individual models on multiple source domains using ERM and subsequently transferring the knowledge from this ensemble (akin to a mixture of experts) to the target domain. Additionally, while the authors introduce a complex method named ' multi-class SML' to determine the aggregation of predictions from the source models, [3] employs a more straightforward, learnable transformer encoder for the same purpose, with parameters learned through meta-learning. Notably, this paper omits citations and comparisons to [3].

* The paper lacks a comprehensive 'related work' section, leading to the omission of some pivotal prior research. For instance, meta-learning, a critical concept for addressing domain generalization, was introduced in [4]. Given that "meta" appears in the paper's title, the authors should clarify how this concept ties into their proposed method, using standard terminology such as 'support' and 'query' sets. Additionally, it would be beneficial for the authors to draw comparisons between their work and other studies that have also employed meta-learning for domain generalization.


[1] A Comprehensive Survey on Test-Time Adaptation under Distribution Shifts. 2023

[2] Multi-Source Domain Adaptation with Mixture of Experts. EMNLP 2018

[3] Meta-DMoE: Adapting to Domain Shift by Meta-Distillation from Mixture-of-Experts. NeurIPS 2022

[4] Adaptive Risk Minimization: Learning to Adapt to Domain Shift. NeurIPS 2021

**Questions:**

* Please make a comparison with [3] and identify the differences in detail. At the conceptual level, they are the same. Specifically, they both model source domain knowledge using the ensemble of models and softly determine the contribution of each source domain to the target. At the technical level, the paper proposes a different method to aggregate the source knowledge in the ensemble with an additional  "meta-performance estimation" to select the source models within the ensemble.

* Please clarify how the concept of "meta" relates to the proposed approach. Is it the same with "meta-learning"? If yes, please explain what are the support set and query set. What is the meta-knowledge is learned during meta-training?

* Please provide more explanation of the design of "Meta model selection", especially from the perspective of intuition. It is a little bit confusing with what is described in the paper "In this way, the instability of stochastic optimization can be accommodated,"

* Please add a related work section.

---

> ### Author Response · Authors · 2023-11-17
> **Rebuttal-Reviewer 52us-(1/3)**
>
> Thank you for your constructive comments and suggestions, and they are exceedingly helpful for us to improve our paper. We have carefully incorporated them in the revised paper. In the following, your comments are first stated and then followed by our point-by-point responses.
>
> *1. The research lacks novelty in its claim. The problem addressed in this paper, namely Test-time adaptation for Distribution shifts, was comprehensively discussed in [1]. Furthermore, the approach of aggregating knowledge from multiple source domains to an unseen target domain, often using techniques like Mixture-of-Experts or ensemble, is not a novel concept either [2, 3].*
>
> Response: Thank you for your careful examination. Firstly, we want to clarify that we are fully aware of the test-time adaptation (TTA) [1] literature. Secondly, we appreciate your highlighting two important papers [2,3], namely the Mixture-of-Experts (MoE) models for DG.
>
> Regarding TTA, [1] was explicitly cited twice in our initial submission. In the Introduction, we mentioned, "recent advances in test-time adaptation (Liang et al., 2023) shed light on the feasibility of utilizing information from test data in real-time". Additionally, In Appendix A.1.2, we stated, “Considering further real-time or online adaptation during the test phase, test-time adaptation (Liang et al., 2023) approaches could also be utilized in DG setting based on the specific application requirements. Such approaches usually need to further adapt the model parameters or introduce extra structures into the framework, and are not the focus of this work.” Reviewer ExFD noted this distinction, emphasizing that “SMEE entails no model re-training, adaptation, or iterative optimization, making it amenable to online incremental settings where test samples arrive sequentially.”
>
> To elaborate further, let's examine currently available/popular TTA approaches for illustration. Uncertainty-based approaches involve additional optimization, as seen in Tent [5], DELTA [6], MEMO [7], SAR [8]. These methods update model parameters at test-time using test prediction entropy, with further assumptions based on normalization statistics, class ratio, label distribution, augmentations, loss-sharpness, respectively. Data-based approaches use pseudo labels for distance-based classification, as in T3A [9] and TAST [10], or employ a memory bank from teacher-student distillation, like CoTTA [11] and RoTTA [12].
>
> In contrast, our approach computes the weights for the source experts ensemble directly using matrix operations. It is important to note that SMEE and current TTA approaches provide entirely different solutions, even though they consider an identical setting.
>
> Regarding the MoE and meta-learning, the response follows the subsequent question.
>
> *2. Conceptually, the research presented in this paper bears a significant resemblance to [3]. Both works utilize a similar approach, pretraining individual models on multiple source domains using ERM and subsequently transferring the knowledge from this ensemble (akin to a mixture of experts) to the target domain. Additionally, while the authors introduce a complex method named ' multi-class SML' to determine the aggregation of predictions from the source models, [3] employs a more straightforward, learnable transformer encoder for the same purpose, with parameters learned through meta-learning. Notably, this paper omits citations and comparisons to [3].*
>
> *Question 1: Please make a comparison with [3] and identify the differences in detail. At the conceptual level, they are the same. Specifically, they both model source domain knowledge using the ensemble of models and softly determine the contribution of each source domain to the target. At the technical level, the paper proposes a different method to aggregate the source knowledge in the ensemble with an additional "meta-performance estimation" to select the source models within the ensemble.*
>
> Response: Thank you for the insightful comment. Our terminologies we used overlap with "meta"-learning and mixture-of-"experts" approaches, although they refer to completely different techniques. A similar concern was also raised by reviewer X4Bt.
>
> To address this, we explain the core concepts of SMEE below. Let us first decompose our proposed SMEE and Meta-DMoE into components.

---

> ### Author Response · Authors · 2023-11-17
> **Rebuttal-Reviewer 52us-(2/3)**
>
> The steps of SMEE are as follows:
>
> 1. **Train domain-specific experts**: For each source domain, we train multiple experts with different random seeds. Each expert is optimized on a single source domain.
>
> 2. **Improve experts using meta model selection**: We directly evaluate each expert’s performance (accuracy scores) on other source domains and save those with higher scores. Each expert is now optimized on a single source domain but improved by knowledge from other source domains.
>
> 3. **Predict test samples from the target domain in online incremental testing**: use expert prediction probabilities to calculate the covariance matrix up to that specific test sample to decide the ensemble weights of experts.
>
> The steps of Meta-DMoE are as follows:
>
> 1. **Train domain-specific experts**: Each expert is optimized on a single source domain.
>
> 2. **Improve experts using meta-learning under knowledge distillation**: Split the data in each source domain into support and query sets. Use the unlabeled support set for unsupervised adaptation via knowledge distillation in the inner loop to obtain the adapted feature extractor. The labeled query set is used to evaluate the adapted feature extractor to perform a meta-update. Each expert is now optimized on a single source domain but improved by knowledge from other source domains.
>
> 3. **Sample an unlabeled support set from the target domain**: Use this support set to obtain an updated model.
>
> 4. **Predict the rest of the test samples from the target domain**.
>
> We emphasize the following distinctions:
>
> 1. **The "meta" components in SMEE have different meanings than those in "meta-learning."** "Meta" model selection refers to cross-validation without accessing the test set, and in Spectral Meta-Learner, "meta" means "ensemble" without being supervised, unrelated to the gradient-based Model-Agnostic Meta-Learning (MAML) [13] or few-shot learning paradigm of support/query set splits.
>
> 2. In SMEE, ensemble weights are calculated after a Softmax operation on **prediction probabilities** for performance estimation, while in Meta-DMoE (and most other MoEs), ensemble weights are calculated in the **feature space** using distance metrics, and then converted to probabilities.
>
> 3. Ensemble weights are **learned through optimization** in MoE, whereas in SMEE, ensemble weights are **directly calculated** without any learnable parameters or hyperparameters.
>
> 4. SMEE assigns **different ensemble weights for each test sample** in online incremental testing, fitting with this scenario, which differs from intuition of all meta-learning and MoE approaches.
>
> 5. SMEE **does not optimize/retrain** the model on the target domain, contrasting with Meta-DMoE and other approaches that use unlabeled support sets from the target domain for offline updates. For Meta-DMoE, an unlabeled support set of size 64 from the target domain (which is supposed to be kept as the test set in DG setting) is used to update the model. [2] uses the full unlabeled target domain offline. Meta-learning such as [4] randomly sample a batch of unlabeled samples from the target domain to obtain an updated model and final predictions, and meta-learning such as [4] also considers single-model DG.
>
> We make the comparison in the form of table for clarity:
>
> | algorithm | target samples needed? | further optimization on target? |  experts are domain-specific? | where was aggregation done? |
> | -------- | ------- | ------- | ------- | ------- |
> | SMEE | Online incremental testing | No | Yes | prediction probabilities |
> | Meta-DMoE [3] | Yes, a batch | Yes | Yes | feature space (transformer as aggregator) |
> | GMoE [14] | No | No | No | feature space (top-k cosine router) |
> | MoDE [15] | No | No | Yes | feature space (MLP as gating network) |
> | RaMoE [16] | No | No | Yes | feature space (similarity between prototypes as weights) |
> | [2] | Yes, all target | Yes, offline combined optimization | Yes | feature space (point-to-set Mahalanobis distance) |
> | ARM [4] | Yes, a batch | Yes | No, single-model | / |
>
> For empirical comparison, we compared with Meta-DMoE and GMoE [14] on DomainBed benchmark. The results are shown in Table 1 of the paper, and we also present them below for your easy reference:
>
> | Algorithm | VLCS | PACS | OfficeHome | TerraIncognita | DomainNet | Average |
> | -------- | ------- | ------- | ------- | ------- | ------- | ------- |
> | Meta-DMoE | / | 86.9 | / | / | 44.2 | / |
> | GMoE  | 80.2$_{\pm0.2}$ | 88.1$_{\pm0.1}$ | **74.2**$_{\pm0.4}$ | 48.5$_{\pm0.1}$ | **48.7**$_{\pm0.2}$ | 67.9 |
> | SMEE (ours) | **82.1**$_{\pm0.4}$ | **90.4**$_{\pm0.4}$ | 73.2$_{\pm0.6}$ | **52.8**$_{\pm2.1}$ | 48.1$_{\pm0.6}$ | **69.3** |
>
> By now, it should be easy to discern the differences at both conceptual and empirical levels. The approaches barely share any similarities, except for using the terms "meta" and "expert." We also argue that our approach is far from complex. In practice it only yields computing the SVD, as in the pseudocode.

---

> ### Author Response · Authors · 2023-11-17
> **Rebuttal-Reviewer 52us-(3/3)**
>
> 3. *Please provide more explanation of the design of "Meta model selection", especially from the perspective of intuition. It is a little bit confusing with what is described in the paper "In this way, the instability of stochastic optimization can be accommodated."*
>
> Response: Thank you for your thorough examination. We elaborate on the intuition and details below.
>
> The optimization process of stochastic mini-batch gradient descent is unstable. As illustrated in Figure 3 and Figure 6 using PACS dataset in our paper, test performance can vary significantly, up to 10%, depending on the validation set choice or early stopping strategy.
>
> DomainBed has emphasized the extreme importance of the validation strategy in a DG setting, as quoted in our paper: "A DG algorithm should be responsible for specifying a model selection method", in Section 3.3. Specifically, the training-domain validation strategy, which involves splitting and combining 20% of the training data from each source domain as the validation set, is no longer optimal.
>
> Why is the training-domain validation not optimal under the domain-specific experts paradigm? The purpose of a validation set is to assess the performance of a model trained on the training set. In DG, the goal is to maximize the out-of-distribution generalization ability of such performance. Therefore, ideally, the validation set should have a different distribution than the training set. However, the randomly leave-one-out validation strategy was not optimal, as demonstrated in DomainBed, since randomly choosing such a validation set from the training domains might not necessarily reflect the target domain distribution.
>
> In the case of domain-specific experts, however, we now have access to other LABELED source domains, in contrast to the single-model DG setting where we have no left-out domain for cross-validation. Therefore, we propose a meta model selection that directly evaluates the expert's performance on other source domains. This involves cross-validation without accessing the test set, hence the term "meta." Our approach is simple and intuitive, demonstrating outstanding performance in practice, as shown in Figure 4.
>
> 4. *The paper lacks a comprehensive 'related work' section, leading to the omission of some pivotal prior research. For instance, meta-learning, a critical concept for addressing domain generalization, was introduced in [4]. Given that "meta" appears in the paper's title, the authors should clarify how this concept ties into their proposed method, using standard terminology such as 'support' and 'query' sets. Additionally, it would be beneficial for the authors to draw comparisons between their work and other studies that have also employed meta-learning for domain generalization.*
>
>     *From Question: Please add a related work section.*
>
> Response: The related work was included in our initial submission at the beginning of the Appendix (Section A.1 Related Work), a common practice in ICLR papers due to page limits. While we acknowledge that reading the appendix is not mandatory during the review process, we strongly encourage reviewers and readers to take a brief look. All the details of the proposed approach are thoroughly explained, addressing potential questions.
>
> We have added a subsection in the related work discussing MoE and meta-learning, hopefully to clear possible confusions from readers who are more familiar with literature on those two topics.
>
> We appreciate your time reviewing the paper, and thank you for highlighting the concerns!
>
> **References**
>
> [1] A Comprehensive Survey on Test-Time Adaptation under Distribution Shifts. 2023.
>
> [2] Multi-Source Domain Adaptation with Mixture of Experts. EMNLP 2018.
>
> [3] Meta-DMoE: Adapting to Domain Shift by Meta-Distillation from Mixture-of-Experts. NeurIPS 2022.
>
> [4] Adaptive Risk Minimization: Learning to Adapt to Domain Shift. NeurIPS 2021.
>
> [5] Tent: Fully Test-Time Adaptation by Entropy Minimization. ICLR 2021.
>
> [6] DELTA: degradation-free fully test-time adaptation. ICLR 2023.
>
> [7] MEMO: Test Time Robustness via Adaptation and Augmentation. NeurIPS 2022.
>
> [8] Towards Stable Test-time Adaptation in Dynamic Wild World. ICLR 2023.
>
> [9] Test-time classifier adjustment module for model-agnostic domain generalization. NeurIPS 2021.
>
> [10] Test-Time Adaptation via Self-Training with Nearest Neighbor Information. ICLR 2023.
>
> [11] Continual test-time domain adaptation. CVPR 2022.
>
> [12] Robust test-time adaptation in dynamic scenarios. CVPR 2023.
>
> [13] Model-agnostic meta-learning for fast adaptation of deep networks. ICML 2017.
>
> [14] Sparse Mixture-of-Experts are Domain Generalizable Learners. ICLR 2023.
>
> [15] Learning mixture of domain-specific experts via disentangled factors for autonomous driving. AAAI 2022.
>
> [16] Generalizable person re-identification with relevance-aware mixture of experts. CVPR 2021.

---

> > ### Comment · Reviewer_52us · 2023-11-22
> > **Response to author's rebuttal (Part 3/3)**
> >
> > I appreciate the authors mentioning the section on Related work in the Appendix. However, given that the novelty of this paper is primarily at the technical level, the absence of related work in the main body could lead to misunderstandings among readers. For instance, Reviewer ExFD seems to have misconceived the claim of prioritizing domain specificity over generalization as a novel contribution of this work. To avoid such confusion, I suggest incorporating descriptions of test-time adaptation and its approaches for addressing domain shifts in the related work section. It's crucial to highlight differences between the proposed method and previous works, both conceptually and technically. If page limitations constrain the extent of the related works section, consider condensing Section 2. Although the experimental results and figures in Section 2 are informative, the insights on the limitations of single-model domain generalization and the necessity for knowledge transfer from a mixture of experts have been already discussed in prior literature.
> >
> >
> > **My final score to AC:**
> > After reviewing the author's rebuttal, it's evident that the paper aims to tackle a well-established problem, namely TTA for distribution shift, adhering to an existing conceptual paradigm, with its novelty primarily at the technical level. The additional experimental results and explanations have highlighted some advantages of the proposed approach in real-world deployment. However, the paper currently lacks a comprehensive related work section in the main body that clearly outlines the relationship between previous works and the proposed approach. This omission could lead to confusion about the paper's real contributions. Therefore, while I consider improving my score from a strong reject, it still remains below the threshold for acceptance.

---

> ### Comment · Reviewer_52us · 2023-11-22
> **Response to author's rebuttal (Part 1/3)**
>
> Thank you for the author's response that recognizes the proposed method SMEE is one approach for a well-established problem, Test-time adaptation(TTA) for domain shifts [1]. Besides, I believe there's a misunderstanding regarding my initial question. My concern primarily revolves around the novelty of the proposed paradigm to address domain shifts, rather than a direct comparison of the specific approach presented in this paper with those in [1].
>
> In this paper, the authors challenge the prevailing domain generalization (DG) paradigm, which assumes a single model trained on multiple source domains can generalize well across various target domains without access to the target data. The authors argue that this paradigm is sub-optimal and instead propose a paradigm that prioritizes specificity over generalization, detailed described in Section 2. Specifically, the proposed paradigm introduces three changes: Firstly, it emphasizes the domain specialty of the target domain by utilizing unlabeled target data at test time. Secondly, it proposes using multiple individual models to capture the knowledge of multi-source domains because each source domain may contain much domain-specific information. Lastly, it proposes to determine the contribution of each domain-specific model to the unseen target domain based on the target data collected at test-time, rather than uniformly treating the knowledge from all source models as equal.
>
> However, the paradigm with these three changes is not novel. The survey paper [1] categorizes TTA for domain shifts, which allows unlabeled test data from the target domain to be leveraged for domain specialty. Specifically, ARM [2] adapts a single source model at test time to domain shifts using unlabeled data. Following this, Meta-DMoE [3] extends the idea from a single model to multiple source models. This extension enables to model domain-specific knowledge and employs a transformer encoder to determine the contribution of each source model given the target data. This leads to aggregating the multi-source knowledge to the target domain and thus can even handle more complex scenarios like imbalance in categories and data instances [4].
>
> It's important to note that “adaptation” in Test-Time Adaptation (TTA) does not inherently mean optimization or retraining. Rather, it refers to modifying the model's parameters or feature maps in response to test data. Crucially, some techniques facilitate this adaptation with just a single forward pass of the NN. This process can modulate parameters or generate features without the need for computationally intensive optimization methods involving gradient backpropagation.
>
> [1] A Comprehensive Survey on Test-Time Adaptation under Distribution Shifts. 2023
>
> [2] Adaptive Risk Minimization: Learning to Adapt to Domain Shift. NeurIPS 2021
>
> [3] Meta-DMoE: Adapting to Domain Shift by Meta-Distillation from Mixture-of-Experts. NeurIPS 2022
>
> [4] WILDS: A Benchmark of in-the-Wild Distribution Shifts. ICML 2021

---

> ### Comment · Reviewer_52us · 2023-11-22
> **Response to author's rebuttal (Part 2/3)**
>
> Thank you for the detailed clarification and comparison between the proposed method, SMEE, and Meta-DMoE [3]. This comparison effectively highlights the novel aspects of the paper at the technical level, particularly emphasizing its empirical results and computational efficiency. However, I remain unconvinced about the conceptual novelty of the proposed approach. Additionally, it is important to note that the summary and description of Meta-DMoE provided in your rebuttal are inaccurate. Firstly, Meta-DMoE does not utilize meta-learning to enhance expert models. Furthermore, the ensemble weights of each expert model at test time are directly computed via a single forward pass through the transformer encoder without optimization. Moreover, the proposed ensembling approach falls under the mixture-of-experts (MoE) framework. In MoE, each model acts as an expert for a specific region of the input space, and a gating function determines the aggregation of these expert models [5, 6]. Conceptually, it is irrelevant whether this process of aggregation occurs in feature space or output space.
>
> To assist both the author and the reviewers in understanding the concepts presented in these two works, I will borrow a part of the author’s response and explain the similarities and differences between them in a more intuitive way.
>
> In brief, while they diverge at the technical level, both approaches are conceptually similar.  Conceptually, both papers focus on the problem of test-time adaptation for distribution shift in a multi-domain scenario. To tackle this problem, they both follow a two-step process:
> * **Modeling Multi-Source Domains with Mixture-of-experts**: Rather than relying on a single model to represent multiple source domains, both works train individual domain-specific models. These models are treated as a mixture of experts (MoE).
> * **Knowledge Transfer Mechanism**: Upon encountering unlabeled target data at test-time, both approaches propose the mechanism to transfer knowledge from the multiple domain experts to the target. This includes determining which domain-expert model should contribute to the knowledge aggregation and quantifying the extent of its contribution.
>
> Technically, the primary distinction between the two approaches lies in the second step: the selection of domain experts in response to the target data. To enhance out-of-distribution performance, SEMM first introduces a unique strategy, named meta-model selection that aims to filter out domain-specific experts that underperform in other source domains. That can be seen as a heuristic selection mechanism. For each test data, SEMM further utilizes Multi-Class SML to measure the transferability of each expert model’s output by calculating their covariance matrix, leading to the ensemble weights of each expert model.
>
> In contrast, Meta-DMoE approaches the problem differently. It takes unlabeled test data as input to query each expert's output, then uses a transformer block to aggregate these outputs into a pseudo label. The ensemble weights of each expert model are calculated and stored in the attention map of multi-head self-attention layers within the transformer block. This pseudo label then serves as a supervision signal to guide a lightweight student network to adapt to the target domain. To learn the parameter of the transformer block, Meta-DMoE leverages episodic learning, namely meta-learning, where it simulates the out-of-distrbution scenario by excluding the corresponding expert model in each episode.
>
> In summary, the proposed approach focuses on the well-established problem of test-time adaptation for domain shifts in a multi-domain scenario. It adheres to the existing Mixture-of-experts (MoE) paradigm, which emphasizes domain specificity by modeling domain-specific knowledge in individual experts and aggregating this knowledge for the target domain based on target data. The novelty of this paper is limited in proposing a distinct approach that demonstrates effectiveness in empirical results and offers additional advantages for real-world deployment.
>
> Considering the concerns raised by other reviewers regarding computational cost, I recommend that the authors use MACS (Multiply-Accumulates) to measure computational efficiency. Relying solely on computation time in seconds, even for the same program on the same hardware, can yield variable results due to factors such as different versions in PyTorch or low-level kernel implementations, as well as operating system scheduling, like CPU core allocation to other programs.
>
> [5] Mixture of experts: a literature survey. Artificial Intelligence Review 2012
>
> [6] Probabilistic Machine Learning: An Introduction. 2022

---

> ### Author Response · Authors · 2023-11-22
> **Second Rebuttal-Reviewer 52us-(1/2)**
>
> Thank you for your reply and analysis! We appreciate the reply that helps to clarify things.
>
> In the previous rebuttal reply, we have cleared the confusion on details, and now we focus on the contribution part, questioned by reviewer 52us, since Meta-DMoE and ARM approaches consider to address the DG problem from, arguably, a similar way also using multi-expert.
>
> In reply, we emphasize that in our paper, the multi-expert paradigm is what we advocate, not what we originally propose. We never argued or claimed that the paradigm is our original contribution.
>
> As in Abstract:
>
> "This paper departs from the conventional approaches and advocates for a paradigm that prioritizes specificity over broad generalization."
>
> Our main contribution, as in Introduction, never stated that the paradigm itself is our original contribution:
>
> "1. Our work extends the Spectral Meta-Learner (SML) (Parisi et al., 2014) unsupervised ensemble learning approach from binary to multi-class classification, showcasing superior performance over uniform-weights averaging and majority voting with minimal computational overhead.
>
> “ 2. We demonstrate that domain-specific experts in multi-domain learning can be enhanced using a straightforward yet remarkably effective meta model selection approach.”
>
> “ 3. Our proposed test-time transferability estimation and ensemble, leveraging domain-specific experts for multi-domain learning, presents a practical alternative to the prevailing single-model DG paradigm. Comprehensive experiments on DG benchmark datasets illustrate its promise in prioritizing specificity over generalization."
>
> The discussion we offered is for the purpose to point out that this paradigm is more effective, instead of claiming it to be our first innovation. We realize the miscomprehension might be caused by the wording in the writing or missing related work discussion, which was placed in the Appendix. Empirically, we also compared with already existing test-time transferability estimation and ensemble approaches already lightly explored as in the DRM paper. Since the MoE model was not our original consideration, as they are indeed different approaches, as will be detailed again next, the way our paper has presented may have led to the misunderstanding that we tried to make the paradigm our own contribution here. We have improved the writing of the presentation and wording of the paper, to clear the confusion on contribution.
>
> In Introduction, we revised that:
>
> "Inspired by attempts in recent research, we argue that single-model DG is over-optimistic given the wide range of real-world applications. In fact, previous works have already offered enough emphasis on such paradigm with the Mixture-of-Expert (MoE) models~\cite{Zhong2022Meta-DMoE}. "
>
> "In this work, we consider using individual ERM models for each source."  # use "consider" instead of using the word "propose".
>
> In Abstract, we revised that:
>
> "3. Our proposed test-time transferability estimation and ensemble approach follows a more practical alternative to the prevailing single-model DG paradigm."  # use "follows" instead of using the word "presents"
>
> etc.
>
> We would like to thank the reviewers for pointing out the presentation issue. The confusion was not our intention, and we hope to make things clearer.
>
> For distinctions, we have already analyzed and draw the distinctions in previous rebuttal replies, whereas the key distinctions are shown in clear point-to-point comparison. Notably, the key technical distinction with MoE models lies in "where was the aggregation done?" MoE models address the transferability estimation at feature space, not the output space, as in SMEE. The key distinction with ARM is that it uses single-model DG, and SMEE uses domain-specific experts. The essential discrepancy with TTA approach is that there is only performance estimation using the test data stream, without any sort of adaptation to target distribution. Importantly, we reemphasize that our approach directly calculates the model transferability using direct matrix operations, whereas the aforementioned approaches using learnable architectures.

---

> ### Author Response · Authors · 2023-11-22
> **Second Rebuttal-Reviewer 52us-(2/2)**
>
> Here, we explain the SML again using an example, from a more intuitive perspective instead of theoretical formulas, hopefully to draw the distinction with MoE models clearly and close the discussion on similarity with it.
>
> Let us consider applying multiple trained experts to a test data stream. At the beginning of the test stage, we have
> |     |  Test sample 1 |
> | -------- | -------  |
> | Expert 1  | X    |
> | Expert 2 | X    |
> | Expert 3  | O   |
> | Expert 4  | O   |
>
> For test sample 1, we cannot determine whether the prediction should be 'X' or 'O'. However, what if the test data stream continues to query prediction results from the experts?
>
>
> |     |  Test sample 1 | Test sample 2 | Test sample 3 | Test sample 4 | Test sample 5 |
> | -------- | -------  | -------  | -------  | -------  | -------  |
> | Expert 1  | X    |  O  |  X  | X  | X  |
> | Expert 2 | X    |  X  | O  | O  | X  |
> | Expert 3  | O   |  X  | O  | O  | O  |
> | Expert 4  | O   |  X  | O  | O  | O  |
>
> Specifically, for test sample 5, though the predictions are identical to test sample 1, can we make a better ensemble now?
>
> If, in this case, one finds that the prediction for test sample 5 should be 'O', then the SML concept goes through. Expert 3 and 4 share high prediction similarities, and are more likely to be the more accurate experts. Expert 1 has strong disagreement with others, and is likely to be offering false predictions. Therefore, placing more weights on Expert 3 and 4 and less on Expert 1, the final ensemble result should lean towards 'O'. In fact, the four assumptions as in Section 3 are necessary to make such a judgment.
>
> To draw the distinctions, SML decides the ensemble weights through the direct covariance matrix of the experts‘ predictions, while MoE models aim to learn the ensemble weights during training with learnable architectures.
>
> As in Section 3:
>
> "Intuitively, SML utilizes inter-predictor correlations to figure out which predictors make similar and more credible results. "
>
> Our approach, which is (1) supervised performance estimation within the sources + (2) unsupervised performance estimation using test data stream predictions, is essentially different at its core from MoE models or current DG approaches.
>
> We argue again that the technical novelty and original contributions are retained, as stated in the Introduction on contributions.
>
> We thank the reviewer for raising the concerns and giving detailed replies to clarify things. Regarding the last suggestion, we also thank the reviewer for raising the MACS to measure computational cost, which will be incorporated in further revision.

---

### Official Review · Reviewer_X4Bt · 2023-10-31

**Soundness:** 3 good
**Presentation:** 3 good
**Contribution:** 2 fair
**Rating:** 5
**Confidence:** 3

**Summary:**

This paper studies the problem of domain generalization. The proposed method employs domain-specific expert models and leverages un- supervised ensemble learning to create a combination of these experts for better predictions. Experiments are performed on the DomainBed benchmark to show the effectiveness of the proposed method in terms of accuracy and inference efficiency.

**Strengths:**

- The paper introduces an interesting approach to tackle the domain generalization challenge, utilizing an unsupervised ensemble learning technique that improves model selection, with an emphasis on elevating specificity over generalization

- This paper is generally well-structured and easy to follow.

**Weaknesses:**

- The novelty of this paper is somewhat limited, since both mixture-of-experts or spectral meta-learner is not new in this area.

- Some important references are missing, e.g.,
   + [1] Sparse Mixture-of-Experts are Domain Generalizable Learners, ICLR2023.
   + [2] Learning mixture of domain-specific experts via disentangled factors for autonomous driving, AAAI2022
   + [3] Generalizable person re-identification with relevance-aware mixture of experts, CVPR2021

**Questions:**

See weakness.

---

> ### Author Response · Authors · 2023-11-17
> **Rebuttal-Reviewer X4Bt-(1/2)**
>
> Thank you for your constructive comments and suggestions, and they are exceedingly helpful for us to improve our paper. We have carefully incorporated them in the revised paper. In the following, your comments are first stated and then followed by our point-by-point responses.
>
> 1. *The novelty of this paper is somewhat limited, since both mixture-of-experts or spectral meta-learner is not new in this area.*
>
> Response: Thank you for your thorough examination. Regarding mixture-of-experts (MoE), we want to first emphasize that our approach is completely different from MoE, though having same term of "expert". The similar concern and confusion was also raised by reviewer 52us. Let us first decompose our proposed SMEE into components.
>
> The steps of SMEE are as follows:
>
> 1. **Train domain-specific experts**: For each source domain, we train multiple experts with different random seeds. Each expert is optimized on a single source domain.
>
> 2. **Improve experts using meta model selection**: We directly evaluate each expert’s performance (accuracy scores) on other source domains and save those with higher scores. Each expert is now optimized on a single source domain but improved by knowledge from other source domains.
>
> 3. **Predict test samples from the target domain in online incremental testing**: use expert prediction probabilities to calculate the covariance matrix up to that specific test sample to decide the ensemble weights of experts.
>
> We emphasize the following distinctions:
>
> 1. In SMEE, ensemble weights are calculated after a Softmax operation on **prediction probabilities** for performance estimation, while in MoEs, ensemble weights are calculated in the **feature space** using distance metrics, and then converted to probabilities.
>
> 2. Ensemble weights are **learned through optimization** in MoE, whereas in SMEE, ensemble weights are **directly calculated** without any learnable parameters or hyperparameters.
>
> 3. SMEE assigns **different ensemble weights for each test sample** in online incremental testing, fitting with this scenario, which differs from intuition of all meta-learning and MoE approaches.
>
> We make the comparison in the form of table for clarity:
>
> | algorithm | target samples needed? | further optimization on target? |  experts are domain-specific? | where was aggregation done? |
> | -------- | ------- | ------- | ------- | ------- |
> | SMEE | Online incremental testing | No | Yes | prediction probabilities |
> | GMoE [1] | No | No | No | feature space (top-k cosine router) |
> | MoDE [2] | No | No | Yes | feature space (MLP as gating network) |
> | RaMoE [3] | No | No | Yes | feature space (similarity between prototypes as weights) |
> | Meta-DMoE [4] | Yes, a batch | Yes | Yes | feature space (transformer as aggregator) |
> | [5] | Yes, all target | Yes, offline combined optimization | Yes | feature space (point-to-set Mahalanobis distance) |
>
> It should be easier to discern the differences at the conceptual level. The approaches barely share any similarities, except for using the term "expert."
>
> For empirical comparison, we compared with GMoE [1] and Meta-DMoE [4] on DomainBed benchmark. The results are shown in Table 1 of the paper, and we also present them below for your easy reference:
>
> (results to other approaches omitted)
>
> | Algorithm | VLCS | PACS | OfficeHome | TerraIncognita | DomainNet | Average |
> | -------- | ------- | ------- | ------- | ------- | ------- | ------- |
> | Meta-DMoE [4] | / | 86.9 | / | / | 44.2 | / |
> | GMoE [1] | 80.2$_{\pm0.2}$ | 88.1$_{\pm0.1}$ | **74.2**$_{\pm0.4}$ | 48.5$_{\pm0.1}$ | **48.7**$_{\pm0.2}$ | 67.9 |
> | SMEE (ours) | **82.1**$_{\pm0.4}$ | **90.4**$_{\pm0.4}$ | 73.2$_{\pm0.6}$ | **52.8**$_{\pm2.1}$ | 48.1$_{\pm0.6}$ | **69.3** |
>
> Empirically, our approach also performs better.
>
> As for SML in the literature, to our knowledge, SML has not been employed to explicitly address domain shift. Further details are provided in Appendix A.1.3, titled "Unsupervised Ensemble Learning." Specifically, we contend that current approaches are "either evaluated under synthetic or small datasets or have inherent limitations, such as iterative training, extra trainable parameters, and ranking approximation. There still lacks a theoretical and empirical study of SML in the context of contemporary deep learning, which could appropriately address SML's expectation of an infinitely large unlabeled test set, assumption of perfect conditional independence between errors of predictors, and limitation to the binary classification setting."
>
> Our contribution to SML involves extending it to the multi-class case, analyzing its practicability in an online incremental setting to handle multi-experts transfer. We hope this clarifies our contribution in relation to existing literature.

---

> ### Author Response · Authors · 2023-11-17
> **Rebuttal-Reviewer X4Bt-(2/2)**
>
> 2. *Some important references are missing, e.g.,*
>
> *[1] Sparse Mixture-of-Experts are Domain Generalizable Learners, ICLR2023.*
>
> *[2] Learning mixture of domain-specific experts via disentangled factors for autonomous driving, AAAI2022*
>
> *[3] Generalizable person re-identification with relevance-aware mixture of experts, CVPR2021**
>
> Response: Thank you for your thorough examination. In previous question, we extensively discussed the distinctions between MoE and our approach. These methods have few similarities, primarily utilizing the term "expert." Your feedback prompted us to recognize that MoE models are also employed to address domain-shift challenges. In response to your comment, we have introduced a subsection in the related work that discusses MoE, referencing the three papers mentioned above.
>
> "A.1.4 MIXTURE-OF-EXPERTS AND META-LEARNING"
>
> "Mixture-of-Experts (MoE) models, often incorporating meta-learning optimization strategies, have been explored in the DG context. MoE models implicitly learn weights for each respective expert through trainable model parameters, distinguishing them from the explicit weight calculation in SML."
>
> "General MoE approaches typically do not employ domain-specific experts. Generalizable Mixture- of-Experts (GMoE) (Li et al., 2023a) [1] introduces sparse MoEs with a cosine router, emphasizing transformer architecture. Other MoE designs specifically incorporate domain-specific experts. For instance, in offline unsupervised domain adaptation, Guo et al. (2018) proposes an MoE that learns a point-to-set Mahalanobis distance metric to weigh the experts for different target examples, introducing meta-learning within the sources to address the unlabeled target domain. Given the inaccessibility of the target domain in DG, subsequent works often resort to the meta-learning paradigm, using support set and query set splits within the source domains to meta-learn the proper assignment of aggregation weights. Mixture of Domain-specific Experts (MoDE) (Kim et al., 2022) [2], Relevance-aware MoE (RaMoE) (Dai et al., 2021) [3], and Meta-Distillation of MoE (Meta-DMoE) (Zhong et al., 2022) fall under this concept. The choice of distance metrics may include L2-distance, cosine distance, or Mahalanobis distance, depending on the design. Typically, a small amount from the target domain serves as the support set to further optimize such models, as observed in Meta-DMoE or ARM."
>
> "Notably, MoE approaches focus on learning the metric function or ensemble weights of aggregation in the feature space during the training phase, while SML concentrates on analyzing cross-sample correlations in prediction probabilities during the test phase."
>
> We appreciate your careful review and thank you a lot for your suggestion in enhancing the paper!
>
> **References**
>
> [1] Sparse Mixture-of-Experts are Domain Generalizable Learners. ICLR 2023.
>
> [2] Learning mixture of domain-specific experts via disentangled factors for autonomous driving. AAAI 2022
>
> [3] Generalizable person re-identification with relevance-aware mixture of experts. CVPR 2021.
>
> [4] Meta-DMoE: Adapting to Domain Shift by Meta-Distillation from Mixture-of-Experts. NeurIPS 2022.
>
> [5] Multi-Source Domain Adaptation with Mixture of Experts. EMNLP 2018.

---

### Official Review · Reviewer_ExFD · 2023-11-01

**Soundness:** 3 good
**Presentation:** 3 good
**Contribution:** 3 good
**Rating:** 6
**Confidence:** 4

**Summary:**

This paper tackles domain generalization (DG) aiming to construct a uniﬁed model trained on diverse source domains, with the goal of achieving robust performance on any unseen test domain.  This paper proposes using individual ERM models for each source and aggregating their predictions during the test phase, by a meta performance estimation technique for model selection within the sources. Furthermore, an approach based on spectral unsupervised ensemble learning to assess the transferability of each source model to test samples is proposed.

**Strengths:**

+ Observation and Motivation: the finding is quite important, and the motivation is quite clear by showing Figure 2 and Figure 3: the transferability of any source domain remains unpredictable without access to the speciﬁc test domain information, leading to the inherent instability witnessed in current DG approaches. The solution is to use domain-speciﬁc experts to prioritize speciﬁcity over generalization.
+ Methodology: the introduction of a spectral ensemble for source models is both innovative and practical, effectively enhancing the robustness of the DG framework. This approach incurs the additional task of recalculating the covariance matrix with cumulative test set data, but the trade-off is well justified by the benefits.
+ Presentation: the clarity and precision of the writing, complemented by well-crafted figures, make the content not only accessible but also engaging. The overall presentation is of high quality, effectively conveying complex concepts in a coherent manner.
+ Experiments: the experimental design is thorough and multifaceted, convincingly demonstrating the efficacy of the proposed method from the following perspectives: 1) meta model selection before ensemble 2) domainbed benchmark 3) test-time ensemble 4) test-time transferability estimation. The performance is outstanding.

**Weaknesses:**

- Computation Complexity: while the proposed spectral ensemble-based method offers the advantage of being applicable in online incremental settings without the need for re-training, adaptation, or iterative optimization, it still presents certain limitations. Specifically, the need to recalculate the covariance matrix and compute the Singular Value Decomposition (SVD) may result in lower computational efficiency compared to other Domain Generalization (DG) methods that do not require any post-hoc computations, such as MIRO and Fishr. This increased computational overhead could potentially slow down inference speed, posing a challenge for deploying the model in real-world scenarios.

---- After Rebuttal ------
We thank the detailed comparisons on inference time provided and the clarification on pseudo-code. After carefully reading Reviewer 52us's comments, I think I have missed a few relevant works in this domain and comparably the originality of this work is somewhat limited. The main technical component derived is originated from another paper and there are already a few emsemble-based DG methods in the past. But the good side is that the proposed method indeed increases the performance on most of benchmark datasets. Based on the comments above, I would like to change my score to weak accept.

**Questions:**

1. Refer the weakness, I am wondering the actual inference time for each target sample compared with the baselines (MIRO and Fishr).
2. In Algorithm 1, M is the number of total models been kept after meta ranking, However, for meta estimation, “for i = 1 : S” means for each source domain, “Rank and select the best performing M models”. Does it mean each source domain will have M best performing models, so the total number of models becomes S*M. It is confusing here, because M is the predictors are available to make predictions on test samples (Sec 3.2). Could you please clarify the number of models chosen from each source domain, and explain how to get M in both Algorithm 1 and Section 3.3?

---

> ### Author Response · Authors · 2023-11-17
> **Rebuttal-Reviewer ExFD-(1/2)**
>
> Thanks for your encouraging words and constructive comments! We sincerely appreciate your time in reviewing the paper, and our point-to-point responses to your comments are given below.
>
> 1. *Weakness of Computation Complexity: while the proposed spectral ensemble-based method offers the advantage of being applicable in online incremental settings without the need for re-training, adaptation, or iterative optimization, it still presents certain limitations. Specifically, the need to recalculate the covariance matrix and compute the Singular Value Decomposition (SVD) may result in lower computational efficiency compared to other Domain Generalization (DG) methods that do not require any post-hoc computations, such as MIRO and Fishr. This increased computational overhead could potentially slow down inference speed, posing a challenge for deploying the model in real-world scenarios.*
>
> Response: Thank you for the insightful comment! The computational complexity in an online incremental setting is indeed crucial in determining the applicability of the approach to specific real-world scenarios. Reviewer 6gJi also raised the same concern. In our initial submission, we reported the computational cost on each dataset in Appendix A.8, but we acknowledge that it was not comprehensive. Building on your feedback, we have expanded our analysis and discussion on the computation cost, incorporating these improvements into our revision.
>
> The maximum computation cost for each of the dataset used was recorded, on a Intel(R) Xeon(R) Platinum 8176 CPU @ 2.10GHz. The results are shown in Table 4 of the paper, and we also present them below for your easy reference:
>
> | Dataset    |  SVD matrix maximum size (# experts, # samples, # classes) | Computation time (seconds) |
> | -------- | ------- | ------- |
> | VLCS  | (15, 1415, 5) | 0.008   |
> | PACS | (15, 2048, 7) | 0.019    |
> | OfficeHome  | (15, 2427, 65) | 0.535   |
> | TerraIncognita  | (15, 4741, 10) | 0.109   |
> | DomainNet |  (25, 48129, 345) | 61.423   |
>
> Note that the table showed the maximum computation time. In online incremental setting, this is the time required for the very last test sample. Except for DomainNet, a substantially larger dataset, multi-class SML can be executed with minimal computational effort in milliseconds. As an extra note of information, the forward pass of ResNet-50 on a single test sample takes around 0.2 second for that same CPU.
>
> It is important to note that the computation time of multi-class SML scales proportionally with the number of classes. In real-world scenarios, especially for datasets like DomainNet, recalculating the covariance matrix for each new incoming test sample becomes impractical. We discuss how to address this challenge in real-world scenarios below.
>
> In Appendix Section A.6, we analyze the differences between offline and complete online versions of multi-class SML. The difference is whether we can use the full test set information, or only a portion of the test set until this specific test sample for the covariance matrix calculation. The results revealed that the disparity between them is relatively small (almost zero difference for small datasets and 0.5%-2% for larger datasets). Importantly, this disparity primarily occurs at the beginning of testing.
>
> To minimize computation costs for large test set with numerous classes, we propose a straightforward strategy. Initially, during the test phase, the matrix size remains small, facilitating efficient updates of ensemble weights for every test sample. As demonstrated in the table, this process takes only milliseconds. Once a sufficient number of test sample predictions have been accumulated, updating the weights for each new sample would only introduce minor changes to the ensemble weights. Consequently, the update can now be performed after a batch of test queries.
>
> **To summarize**, ideally, the update to ensemble weights should be done for every single test sample at the beginning of the test phase. When we have accumulated enough test sample predictions, this update can be scheduled at intervals, presumably during periods when the system is idle and has abundant computation resources. The exact point of this transition depends on the computational resources available and the actual dataset in practice. To gauge when this transition point should occur, we refer to Figure 9, where we compare offline and online prediction disparities.

---

> ### Author Response · Authors · 2023-11-17
> **Rebuttal-Reviewer ExFD-(2/2)**
>
> 2. *Refer the weakness, I am wondering the actual inference time for each target sample  compared with the baselines (MIRO and Fishr).*
>
> Response: Thank you for the careful examination. We discussed how to minimize the computation time of multi-class SML above, and we delve deeper into this aspect from the perspective of three real-world deployment scenarios.
>
> MIRO and Fishr are **training-time ensemble** approaches. In the end, they transfer a single model to the target domain, maintaining their inference time at test-time at the cost of a single forward pass. In contrast, strategies like DRM and SMEE transfer multiple experts and consider **test-time ensemble**.
>
> We specifically explore **three possible deployment settings** to discuss how the approaches would fit:
>
> 1. **Experts as APIs offered by companies:** These APIs return only prediction results. This is where multi-class SML shines, operating solely on prediction probabilities without requiring access to model parameters, as discussed in Section 3.4. The computation can be performed on the user's end, such as a smartphone, using the strategy discussed above. The primary computation overhead would likely be the communication time of the APIs in this case.
>
> 2. **Experts deployed on the user's end with multiple available CPUs:** In this scenario, parallel computing of the forward passes of respective experts is feasible. The computation overhead would depend on both the matrix operations of multi-class SML and the forward passes on the CPUs. For DRM or SIMPLE, the computation overhead would depend on both their ensemble strategy calculation and the forward passes on the CPUs. The actual time depends on the architectures of models.
>
> 3. **Single CPU executing forward passes of multiple experts:** In cases where only a single CPU is available for executing the forward passes of multiple experts, the test-ensemble strategy operates slowly. Here, the computation overhead would be (#experts * forward_pass_time + ensemble_cost) for multi-class SML, DRM, and SIMPLE. The computation time of multi-class SML is not vital in this case.
>
> To summarize, the computation time really depends on the actual scenario in practice. Our approach could be applicable to multiple scenarios with specific forms of adaptations, with the strategy discussed in reply to the last question.
>
> 3. *In Algorithm 1, M is the number of total models been kept after meta ranking, However, for meta estimation, “for i = 1 : S” means for each source domain, “Rank and select the best performing M models”. Does it mean each source domain will have M best performing models, so the total number of models becomes S*M. It is confusing here, because M is the predictors are available to make predictions on test samples (Sec 3.2). Could you please clarify the number of models chosen from each source domain, and explain how to get M in both Algorithm 1 and Section 3.3?*
>
> Response: Thank you for the careful examination and for catching the mistake! M represents the total number of models used in the final ensemble (in experiments, M=5 * (# source domains)). We have rectified this in the pseudocode of Algorithm 1. Specifically, M is now defined as "the total number of models kept after meta-ranking from all source domains." In the loop, we have also adjusted it to "Rank and select the best-performing M/S models." After the loop, we add a statement that "Assemble all the selected models."
>
> We appreciate your careful review and thank you for your time and efforts in enhancing the paper!

---

### Meta-Review · Area_Chair_XCvk · 2023-12-07

**Metareview:**

This paper proposes Spectral Meta Estimation and Ensemble (SMEE), a method for making predictions in a test domain by combining outputs of models trained on source domains. Technically, SMMEE is an extension of Spectral Meta-Learner (SML) (Parisi et al., 2014) from binary to multi-class classification. (SML has so far had limited impact judging from the citation number.)  Reviewer 52us has engaged in a long discussion with the authors and remained unconvinced of the novelty of the work.  Reviewer ExFD was originally very positive (giving a 8), but became less supportive after the discussion.  In fact, he/she wrote: “after carefully reading Reviewer 52us's comments, I think … the originality of this work is somewhat limited.”  Reviewer X4Bt also think “the novelty of this paper is somewhat limited”.  The AC agrees with them.

**Justification For Why Not Higher Score:**

See above

**Justification For Why Not Lower Score:**

See above

---

### Decision · Program_Chairs · 2024-01-16

Reject